# ASYNCHRONOUS MULTI-AGENT ACTOR-CRITIC WITH MACRO-ACTIONS

## ABSTRACT

Many realistic multi-agent problems naturally require agents to be capable of performing asynchronously without waiting for other agents to terminate (e.g., multi-robot domains). Such problems can be modeled as Macro-Action Decentralized Partially Observable Markov Decision Processes (MacDec-POMDPs). Current policy gradient methods are not applicable to agents' asynchronous decision-making over macor-actions in MacDec-POMDPs, as these methods assume that agents synchronously reason about action selection at every timestep. To allow asynchronous learning and decision-making, we formulate a set of asynchronous multi-agent actor-critic methods that allow agents to directly optimize asynchronous (macro-action-based) policies in three standard training paradigms: decentralized learning, centralized learning, and centralized training for decentralized execution. Empirical results in various domains show high-quality solutions can be learned for large domains when using our methods.

## 1 INTRODUCTION

In recent years, multi-agent policy gradient methods using the actor-critic framework have achieved impressive success in solving a variety of cooperative and competitive domains (Lowe et al., 2017; Foerster et al., 2018; Du et al., 2019; Iqbal & Sha, 2019; Vinyals et al., 2019; Li et al., 2019; Wang et al., 2020; Yang et al., 2020; Zhou et al., 2020; Baker et al., 2020; Su et al., 2021; Wang et al., 2021; Du et al., 2021). However, as these methods assume synchronized primitive-action execution over agents, they struggle to solve tasks that involve long-term reasoning and asynchronous behavior, such as real-world multi-robot applications (e.g., search and rescue (Queralta et al., 2020), package delivery (Choudhury et al., 2021) and warehouse service (Xiao et al., 2020)).

The *Macro-Action Decentralized Partially Observable Markov Decision Process* (MacDec-POMDP) (Amato et al., 2014; 2019) provides a general formalism for multi-agent asynchronous collaborative decision-making under uncertainty. Macro-actions represent temporally extended actions that have (potentially) different durations. This introduces asynchronous high-level decision-making over agents, as agents can start and terminate macro-actions at different timesteps. Such asynchronicity actually makes multi-agent reinforcement learning (MARL) more challenging because it is difficult to determine what information to use and when to update agents' policies from either the decentralized or centralized perspective.

Despite several efforts made recently to enable agents to learn asynchronous hierarchical policies such as extending DQN (Mnih et al., 2015) to learn macro-action-value functions (Xiao et al., 2019), transferring MacDec-POMDPs to event-driven processes with continuous timing (Menda et al., 2019), and adapting a single-agent *option-critic* framework (Bacon et al., 2017) to multi-agent domains to learn all components (e.g. low-level policy, high-level abstraction, high-level policy) from scratch (Chakravorty et al., 2019), none of them provides a principled way for optimizing macro-action-based policies via asynchronous policy gradients to solve general multi-agent problems with asynchronous decision-making.

In this paper, we propose a group of macro-action-based multi-agent actor-critic methods to generalize the current primitive-action-based multi-agent actor-critic methods to multi-agent problems with macro-actions as well as allowing asynchronous policy optimization. First, we formulate a *macro-action-based independent actor-critic* (Mac-IAC) method. Although independent learning suffers from a theoretical curse of environmental non-stationarity, it allows fully online learning and may

still work well in certain domains. Second, we introduce a *macro-action-based centralized actor-critic* (Mac-CAC) method, for the case where full communication is available during execution. We also formulate a centralized training for decentralized execution (CTDE) paradigm (Kraemer & Banerjee, 2016; Oliehoek et al., 2008) variant of our method. CTDE has gained popularity since such methods can learn better decentralized policies by using centralized information during training. Current primitive-action-based multi-agent actor-critic methods typically use a centralized critic to optimize each decentralized actor. However, the asynchronous joint macro-action execution from the centralized perspective could be very different with the completion time being very different from each agent's decentralized perspective. To this end, we first present a *Naive Independent Actor with Centralized Critic* (Naive IACC) method that naively uses a joint macro-action-value function as the critic for each actor's policy gradient estimation; and then propose an *Independent Actor with Individual Centralized Critic* (Mac-IAICC) method addressing the above challenge.

We evaluate our proposed methods on diverse macro-action-based multi-agent problems: a benchmark Box Pushing domain (Xiao et al., 2019), a variant of the Overcooked domain (Wu et al., 2021) and a larger warehouse service domain (Xiao et al., 2019). Experimental results show that our methods are able to learn high-quality solutions while primitive-action-based methods cannot, and show the strength of Mac-IAICC for learning decentralized policies over Naive IAICC and Mac-IAC. To our knowledge, this is the first general formalization of macro-action-based multi-agent actor-critic frameworks considering the three state-of-the-art multi-agent training paradigms.

## 2 BACKGROUND

This section first introduces the formal definitions of the Dec-POMDP and the MacDec-POMDP, and then reviews single-agent and multi-agent actor-critic policy gradient methods with primitive-actions. We also provide an overview of value-based MARL methods with macro-actions.

### 2.1 DEC-POMDPs AND MACDEC-POMDPs

The decentralized partially observable Markov decision processes (Dec-POMDP) (Oliehoek & Amato, 2016) is a general framework to model fully cooperative multi-agent tasks, where agents make decisions in a decentralized way based on only local information. Formally, a Dec-POMDP is defined by a tuple $\langle I, S, A, \Omega, T, O, R, \mathbb{H}, \gamma \rangle$, where $I$ is a set of agents; $S$ is the environmental state space; $A = \times_{i \in I} A_i$ is the joint primitive-action space over each agent's primitive-action set $A_i$; $\Omega = \times_{i \in I} \Omega_i$ is the joint primitive-observation space over each agent's primitive-observation set $\Omega_i$. At every timestep, under a state $s$, agents synchronously execute a joint primitive-action $\vec{a} = \times_{i \in I} a_i$, each individually selected by an agent using a policy $\pi_i : H_i^A \times A_i \rightarrow [0, 1]$, a mapping from local primitive observation-action history $H_i^A$ to primitive-actions. The environment then transits to a new state $s'$ according to a state transition function $T(s, \vec{a}, s') = P(s' \mid s, \vec{a})$. Agents receive a global reward $r(s, \vec{a})$ issued by a reward function $R : S \times A \rightarrow \mathbb{R}$, and obtain a joint primitive-observation $\vec{o} = \times_{i \in I} o_i$ drawn from an observation function $O(\vec{o}, \vec{a}, s') = P(\vec{o} \mid \vec{a}, s')$ in state $s'$. The objective is to find a joint policy $\vec{\pi} = \times_i \pi_i$ such that the expected sum of discounted rewards from an initial state, $V^{\vec{\pi}}(s_{(0)}) = \mathbb{E}\left[ \sum_{t=0}^{\mathbb{H}-1} \gamma^t r\left(s_{(t)}, \vec{a}_{(t)}\right) \mid s_{(0)}, \vec{\pi} \right]$, gets optimized, where $\gamma \in [0, 1]$ is a discount factor, and $\mathbb{H}$ is the number of (primitive) timesteps until the problem terminates (the horizon).

The macro-action decentralized partially observable Markov decision process (MacDec-POMDP) (Amato et al., 2014; 2019) incorporates the *option* framework (Sutton et al., 1999) into the Dec-POMDP by defining each agent's macro-action as a tuple $m_i = \langle I_{m_i}, \pi_{m_i}, \beta_{m_i} \rangle$, where the initiation set $I_{m_i} \subset H_i^M$ defines how to initiate a macro-action based on macro-observation-action history $H_i^M$ at the high-level; $\pi_{m_i} : H_i^A \times A_i \rightarrow [0, 1]$ is the low-level policy for the execution of a macro-action; and a stochastic termination function $\beta_{m_i} : H_i^A \rightarrow [0, 1]$ determines how to terminate a macro-action based on primitive-observation-action history $H_i^A$ at the low-level. A MacDec-POMDP is thus formally defined by a tuple $\langle I, S, A, M, \Omega, \zeta, T, O, Z, R, \mathbb{H}, \gamma \rangle$, where $I, S, A, \Omega, T, O, R, \mathbb{H}$ and $\gamma$ remain the same definitions as in the Dec-POMDP; $M = \times_{i \in I} M_i$ is the joint macro-action space over each agent's macro-action space $M_i$; $\zeta = \times_{i \in I} \zeta_i$ is the joint macro-observation space over each agent's macro-observation space $\zeta_i$; and $Z = \{Z_i\}_{i \in I}$ is a set of macro-observation likelihood models. During execution, each agent independently selects a macro-action $m_i$ using a high-level policy $\Psi_i : H_i^M \times M_i \rightarrow [0, 1]$, a mapping from macro-observation-action his-

tory to macro-actions, and captures a macro-observation $z_i \in \zeta_i$ according to the macro-observation probability function $Z_i(z_i, m_i, s') = P(z_i \mid m_i, s')$ when the macro-action terminates in a state $s'$. The objective of solving MacDec-POMDPs with finite horizon is to find a joint high-level policy $\vec{\Psi} = \times_{i \in I} \Psi_i$ that maximizes the value, $V^{\vec{\Psi}}(s_{(0)}) = \mathbb{E}\left[ \sum_{t=0}^{\mathbb{H}-1} \gamma^t r\big(s_{(t)}, \vec{a}_{(t)}\big) \mid s_{(0)}, \vec{\pi}, \vec{\Psi} \right]$.

## 2.2 Single-Agent Actor-Critic

In single-agent reinforcement learning, the *policy gradient theorem* (Sutton et al., 2000) formulates a principled way to optimize a parameterized policy $\pi_\theta$ via gradient ascent on the policy's performance defined as $J(\theta) = \mathbb{E}_{\pi_\theta}\big[ \sum_{t=0}^{\infty} \gamma^t r\big(s_{(t)}, a_{(t)}\big) \big]$. In POMDPs, the gradient w.r.t. parameters of a observation-action history-based policy $\pi_\theta(a \mid h)$ is expressed as:

$$\nabla_\theta J(\theta) = \mathbb{E}_{\pi_\theta}\Big[ \nabla_\theta \log \pi_\theta(a \mid h) Q^{\pi_\theta}(h, a) \Big] \tag{1}$$

where, $h$ is often maintained by having a RNN in the policy network (Hausknecht & Stone, 2015).

The actor-critic framework (Konda & Tsitsiklis, 2000) learns an on-policy action-value function $Q_\phi^{\pi_\theta}(h, a)$ (critic) via *temporal-difference* (TD) learning (Sutton, 1988) to approximate the action-value for the policy (actor) updates. Variance reduction is commonly achieved by training a history-value function $V_{\mathbf{w}}^{\pi_\theta}(h)$ and using it as a baseline (Weaver & Tao, 2001) as well as bootstrapping to estimate the action-value. Accordingly, the actor-critic policy gradient can be written as:

$$\nabla_\theta J(\theta) = \mathbb{E}_{\pi_\theta}\Big[ \nabla_\theta \log \pi_\theta(a \mid h)\big(r + \gamma V_{\mathbf{w}}^{\pi_\theta}(h') - V_{\mathbf{w}}^{\pi_\theta}(h)\big) \Big] \tag{2}$$

where, $r$ is the immediate reward received by the agent at the corresponding timestep.

## 2.3 Independent Actor-Critic

The single-agent actor-critic algorithm can be adapted to multi-agent problems in a simple way such that each agent independently learns its own actor and critic while treating other agents as part of the world (Foerster et al., 2018). We consider a variance reduction version of *independent actor-critic* (IAC) with the policy gradient as follows:

$$\nabla_{\theta_i} J(\theta_i) = \mathbb{E}_{\vec{\pi}_{\vec{\theta}}}\Big[ \nabla_{\theta_i} \log \pi_{\theta_i}(a_i \mid h_i)\big(r + \gamma V_{\mathbf{w}_i}^{\pi_{\theta_i}}(h_i') - V_{\mathbf{w}_i}^{\pi_{\theta_i}}(h_i)\big) \Big] \tag{3}$$

where, $r$ is a shared reward over agents at every timestep. Due to other agents' policy updating and exploring, from any agent's local perspective, the environment appears non-stationary which can lead to unstable learning of the critic without convergence guarantees (Lowe et al., 2017). This instability often prevents IAC from learning high-quality cooperative policies.

## 2.4 Independent Actor with Centralized Critic

To address the above difficulties existing in independent learning approaches, centralized training for decentralized execution (CTDE) provides agents with access to global information during offline training while allowing agents to rely on only local information during online decentralized execution. Typically, the key idea of exploiting CTDE with actor-critic is to train a joint action-value function, $Q_\phi^{\vec{\pi}_{\vec{\theta}}}(\mathbf{x}, \vec{a})$, as the centralized critic and use it to compute gradients w.r.t. the parameters of each decentralized policy (Foerster et al., 2018; Lowe et al., 2017), which can be formulated as:

$$\nabla_{\theta_i} J(\theta_i) = \mathbb{E}_{\vec{\pi}_{\vec{\theta}}}\Big[ \nabla_{\theta_i} \log \pi_{\theta_i}(a_i \mid h_i) Q_\phi^{\vec{\pi}_{\vec{\theta}}}(\mathbf{x}, \vec{a}) \Big] \tag{4}$$

where, $\mathbf{x}$ represents the available centralized information (e.g., joint observation, joint observation-action history, or the true state). Although the centralized critic in Eq. 4 can facilitate the update of decentralized policies in the direction that optimizes global collaborative performance, it also introduces extra variance over other agents' actions (Lyu et al., 2021; Wang et al., 2021). Therefore, we consider the version of *independent actor with centralized critic* (IACC) with a general variance reduction trick (Foerster et al., 2018; Su et al., 2021), the policy gradient of which is:

$$\nabla_{\theta_i} J(\theta_i) = \mathbb{E}_{\vec{\pi}_{\vec{\theta}}}\Big[ \nabla_{\theta_i} \log \pi_{\theta_i}(a_i \mid h_i)\big(r + \gamma V_{\mathbf{w}}^{\vec{\pi}_{\vec{\theta}}}(\mathbf{x}') - V_{\mathbf{w}}^{\vec{\pi}_{\vec{\theta}}}(\mathbf{x})\big) \Big] \tag{5}$$

## 2.5 Learning Macro-Action-Based Deep Q-Nets

An essential aspect of macro-action-based multi-agent systems is the asynchronicity of macro-action execution over agents, where agents may start and complete their own macro-actions at different timesteps. Previous deep MARL methods for Dec-POMDPs cannot work in this case as they are all based on primitive actions synchronously executed by agents. In recent work, macro-action-based multi-agent deep Q-learning methods have been proposed for MacDec-POMDPs (Xiao et al., 2019).

For decentralized learning, a new buffer, *Macro-Action Concurrent Experience Reply Trajectories* (Mac-CERTs), is designed for collecting each agent's macro-observation, macro-action, and reward information. In this buffer, the transition experience of each agent $i$ is represented as a tuple $\langle z_i, m_i, z_i', r_i^c \rangle$, where $r_i^c = \sum_{t=t_{m_i}}^{t_{m_i}+\tau_{m_i}-1} \gamma^{t-t_{m_i}} r_{(t)}$ is a cumulative reward of the macro-action taking $\tau_{m_i}$ timesteps to be completed from its beginning timestep $t_{m_i}$. During training, a mini-batch of concurrent sequential experiences is sampled from Mac-CERTs. Each agent independently accesses its own sampled experiences and obtains a 'squeezed' trajectories by removing the transitions in the middle of each macro-action execution, which ends up with a mini-batch of transitions when the corresponding macro-action terminates (i.e., removing time information). Updates for each macro-action-value function $Q_{\phi_i}(h_i, m_i)$ take place only when the agent's macro-action is complete by minimizing a TD loss over the 'squeezed' data. In the centralized learning case, the objective is to learn a joint macro-action-value function $Q_\phi(\vec{h}, \vec{m})$. To this end, the other special buffer called *Macro-Action Joint Experience Replay Trajectories* (Mac-JERTs) is developed for collecting agents' joint transition experience at every timestep and each is represented as a tuple $\langle \vec{z}, \vec{m}, \vec{z}', \vec{r}^c \rangle$, where $\vec{r}^c = \sum_{t=t_{\vec{m}}}^{t_{\vec{m}}+\vec{\tau}_{\vec{m}}-1} \gamma^{t-t_{\vec{m}}} r_t$ is a shared joint cumulative reward from the beginning timestep $t_{\vec{m}}$ of the joint macro-action $\vec{m}$ to its termination, defined as when *any* agent finishes its own macro-action, after $\vec{\tau}_{\vec{m}}$ timesteps. In each training iteration, the joint macro-action-value function is optimized over a mini-batch of 'squeezed' (depending on each joint macro-action termination) sequential joint experiences via TD learning. Other choices for what information to retain are also possible (e.g., the whole sequence of macro-actions or including time to complete) but this squeezing procedure was found to work well.

In our proposed macro-action-based actor-critic methods, we extend the above approaches to train critics on-policy, and the trajectory squeezing is changed variously for each method in order to achieve improved asynchronous macro-action-based policy updates via policy gradient.

## 3 Approach

Multi-agent deep reinforcement learning with asynchronous decision-making and macro-actions is more challenging as it is difficult to determine *when* to update each agent's policy and *what* information to use. Although the macro-action-based deep Q-learning methods (Xiao et al., 2019) (in Section 2.5) give us the base to learn macro-action value functions, they do not directly extend to the policy gradient case, particularly in the case of centralized training for decentralized execution (CTDE). In this section, we propose principled formulations of on-policy macro-action-based multi-agent actor-critic methods for decentralized learning (Section 3.1), centralized learning (Section 3.2), and CTDE (Section 3.3). In each case, we first introduce the version with a Q-value function as the critic and then present the variance reduction version in our implementation. We use $h_i$ to represent an agent's local macro-observation-action history, and $\vec{h}$ to represent the joint history.

### 3.1 Macro-Action-Based Independent Actor-Critic (Mac-IAC)

Similar to the idea of IAC with primitive-actions (Section 2.3), a straightforward extension is to have each agent independently optimize its own macro-action-based policy (actor) using a local macro-action-value function (critic). Hence, we start with deriving a *macro-action-based policy gradient theorem* in Appendix A.1 by incorporating the general Bellman equation for the state values of a macro-action-based policy (Sutton et al., 1999) into the *policy gradient theorem* in MDPs (Sutton et al., 2000), and then extend it to MacDec-POMDPs so that each agent can have the following policy gradient w.r.t. the parameters of its macro-action-based policy $\Psi_{\theta_i}(m_i|h_i)$ as:

$$\nabla_{\theta_i} J(\theta_i) = \mathbb{E}_{\vec{\Psi}_{\vec{\theta}}} \left[ \nabla_{\theta_i} \log \Psi_{\theta_i}(m_i \mid h_i) Q_{\phi_i}^{\Psi_{\theta_i}}(h_i, m_i) \right] \qquad (6)$$

During training, each agent accesses to its own trajectories and squeezes them in the same way as the decentralized case mentioned in Section 2.5 to train the critic $Q_{\phi_i}^{\Psi_{\theta_i}}(h_i, m_i)$ via on-policy TD learning and perform gradient ascent using Eq. 6 to update the policy when the agent's macro-action terminates. In our case, we train a local history value function $V_{\mathbf{w}_i}^{\Psi_{\theta_i}}(h_i)$ as each agent's critic and use it as a baseline to achieve variance reduction. The corresponding policy gradient is as follows:

$$\nabla_{\theta_i} J(\theta_i) = \mathbb{E}_{\vec{\Psi}_{\vec{\theta}}}\left[\nabla_{\theta_i} \log \Psi_{\theta_i}(m_i \mid h_i)\big(r_i^c + \gamma^{\tau_{m_i}} V_{\mathbf{w}_i}^{\Psi_{\theta_i}}(h_i') - V_{\mathbf{w}_i}^{\Psi_{\theta_i}}(h_i)\big)\right] \tag{7}$$

where, the cumulative reward $r_i^c$ is w.r.t. the execution of agent $i$'s macro-action $m_i$.

## 3.2 MACRO-ACTION-BASED CENTRALIZED ACTOR-CRITIC (MAC-CAC)

In the fully centralized learning case, we treat all agents as a single joint agent to learn a centralized actor $\Psi_\theta(\vec{m} \mid \vec{h})$ with a centralized critic $Q_\phi^{\Psi_\theta}(\vec{h}, \vec{m})$, and the policy gradient can be expressed as:

$$\nabla_\theta J(\theta) = \mathbb{E}_{\Psi_\theta}\left[\nabla_\theta \log \Psi_\theta(\vec{m} \mid \vec{h}) Q_\phi^{\Psi_\theta}(\vec{h}, \vec{m})\right] \tag{8}$$

Similarly, in order to achieve low variance optimization for the actor, we learn a centralized history value function $V_{\mathbf{w}}^{\Psi_\theta}(\vec{h})$ by minimizing a TD-error loss over joint trajectories that are squeezed w.r.t. each joint macro-action termination (as long as one of the agents terminates its macro-action, introduced under the centralized case in Section 2.5). Accordingly, the policy's updates are performed when each joint macro-action is completed by ascending the following gradient:

$$\nabla_\theta J(\theta) = \mathbb{E}_{\Psi_\theta}\left[\nabla_\theta \log \Psi_\theta(\vec{m} \mid \vec{h})\big(\vec{r}^c + \gamma^{\vec{\tau}_{\vec{m}}} V_{\mathbf{w}}^{\Psi_\theta}(\vec{h}') - V_{\mathbf{w}}^{\Psi_\theta}(\vec{h})\big)\right] \tag{9}$$

where the cumulative reward $\vec{r}^c$ is w.r.t. the execution of the joint macro-action $\vec{m}$.

## 3.3 MACRO-ACTION-BASED INDEPENDENT ACTOR WITH CENTRALIZED CRITIC (MAC-IACC)

As mentioned earlier, fully centralized learning requires perfect online communication that is often hard to guarantee, and fully decentralized learning suffers from environmental non-stationarity due to agents' changing policies. In order to learn better decentralized macro-action-based policies, in this section, we propose two macro-action-based actor-critic algorithms using the CTDE paradigm. Typically, the difference between a joint macro-action's termination from the centralized perspective and a macro-action's termination from each agent's local perspective gives rise to a new challenge: *what kind of centralized critic to learn and how to use it to optimize decentralized policies under such an asymmetric asynchrony from the two perspectives*, which we mainly investigate below.

### 3.3.1 NAIVE MAC-IACC

A naive way of incorporating macro-actions into a CTDE-based actor-critic framework is to directly adapt the idea of the primitive-action-based IACC (Section 2.4) to have a shared joint macro-action-value function $Q_\phi^{\vec{\Psi}_{\vec{\theta}}}(\mathbf{x}, \vec{m})$ in each agent's decentralized macro-action-based policy gradient as:

$$\nabla_{\theta_i} J(\theta_i) = \mathbb{E}_{\vec{\Psi}_{\vec{\theta}}}\left[\nabla_{\theta_i} \log \Psi_{\theta_i}(m_i \mid h_i) Q_\phi^{\vec{\Psi}_{\vec{\theta}}}(\mathbf{x}, \vec{m})\right] \tag{10}$$

To reduce variance, with a value function $V_{\mathbf{w}}^{\vec{\Psi}_{\vec{\theta}}}(\mathbf{x})$ as the centralized critic, the policy gradient w.r.t. the parameters of each agent's high-level policy can be rewritten as the following format:

$$\nabla_{\theta_i} J(\theta_i) = \mathbb{E}_{\vec{\Psi}_{\vec{\theta}}}\left[\nabla_{\theta_i} \log \Psi_{\theta_i}(m_i \mid h_i)\big(\vec{r}^c + \gamma^{\vec{\tau}_{\vec{m}}} V_{\mathbf{w}}^{\vec{\Psi}_{\vec{\theta}}}(\mathbf{x}') - V_{\mathbf{w}}^{\vec{\Psi}_{\vec{\theta}}}(\mathbf{x})\big)\right] \tag{11}$$

Here, the critic is trained in the fully centralized manner described in Section 3.2 while allowing it to access additional global information (e.g., joint macro-observation-action history, ground truth state or both) represented by the symbol $\mathbf{x}$. However, updates of each agent's policy $\Psi_{\theta_i}(m_i \mid h_i)$ only occur at the agent's own macro-action termination timesteps rather than depending on joint macro-action terminations in the centralized critic training.

### 3.3.2 INDEPENDENT ACTOR WITH INDIVIDUAL CENTRALIZED CRITIC (MAC-IAICC)

Note that naive Mac-IAICC is technically incorrect. The cumulative reward $\vec{r}^c$ in Eq 11 is based on the corresponding joint macro-action's termination that is defined as when *any* agent finishes its own macro-action, which produces two potential issues: a) $\vec{r}^c + \gamma^{\vec{\tau}_{\vec{m}}} V_{\mathbf{w}}^{\vec{\Psi}_{\vec{\theta}}}(\mathbf{x}')$ may not estimate the value of the macro-action $m_i$ well as the reward does not depend on $m_i$'s termination; b) from agent $i$'s perspective, its policy gradient estimation may involve higher variance associated with the asynchronous macro-action terminations of other agents.

To tackle aforementioned issues, we propose to learn a separate centralized critic $V_{\mathbf{w}_i}^{\vec{\Psi}_{\vec{\theta}}}(\mathbf{x}')$ for each agent via TD-learning. In this case, each TD-error for updating $V_{\mathbf{w}_i}^{\vec{\Psi}_{\vec{\theta}}}(\mathbf{x}')$ is computed by using the reward $r_i^c$ that is accumulated purely based on the execution of the agent $i$'s macro-action $m_i$. With this TD-error estimation, each agent's decentralized macro-action-based policy gradient becomes:

$$\nabla_{\theta_i} J(\theta_i) = \mathbb{E}_{\vec{\Psi}_{\vec{\theta}}}\left[\nabla_{\theta_i} \log \Psi_{\theta_i}(m_i \mid h_i)\left(r_i^c + \gamma^{\tau_{m_i}} V_{\mathbf{w}_i}^{\vec{\Psi}_{\vec{\theta}}}(\mathbf{x}') - V_{\mathbf{w}_i}^{\vec{\Psi}_{\vec{\theta}}}(\mathbf{x})\right)\right] \quad (12)$$

Now, from agent $i$'s perspective, $r_i^c + \gamma^{\tau_{m_i}} V_{\mathbf{w}_i}^{\vec{\Psi}_{\vec{\theta}}}(\mathbf{x}')$ is capable of offering a more accurate value prediction for the macro-action $m_i$, since both the reward, $r_i^c$ and the value function $V_{\mathbf{w}_i}^{\vec{\Psi}_{\vec{\theta}}}(\mathbf{x}')$ depend on agent $i$'s macro-action termination. Also, unlike the case in Naive Mac-IACC, other agents' terminations cannot lead to extra noisy estimated rewards w.r.t. $m_i$ anymore so that the variance on policy gradient estimation gets reduced. Then, updates for both the critic and the actor occur when the corresponding agent's macro-action ends as well as taking the advantage of information sharing.

The pseudo code and detailed trajectory squeezing process for each proposed method are presented in Appendix A.2.

## 4 EXPERIMENTS

### 4.1 ENVIRONMENTS

We investigate the performance of our proposed algorithms over a variety of multi-agent problems with macro-actions (Fig. 1): Box Pushing (Xiao et al., 2019), Overcooked (Wu et al., 2021), and a larger Warehouse Tool Delivery (Xiao et al., 2019) domain. Macro-actions in domains are defined by us using prior domain knowledge as they are straightforward in these tasks. We describe the key properties of each domain here and leave more details in Appendix A.3.

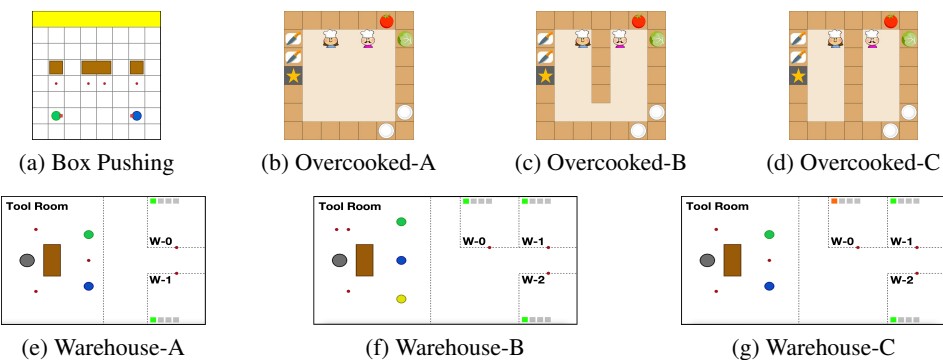

| (a) Box Pushing | (b) Overcooked-A | (c) Overcooked-B | (d) Overcooked-C |

| (e) Warehouse-A | (f) Warehouse-B | (g) Warehouse-C |

Figure 1: Experimental environments.

**Box Pushing** (Fig. 1a). Two agents aim to cooperatively push the big box to the yellow goal area rather than pushing a small box on each own. Boxes can only be moved towards the north. Agents have four primitive-actions: *move forward*, *turn-left*, *turn-right* and *stay*. In the macro-action-based case, each agent has three one-step macro-actions: **Turn-left**, **Turn-right**, and **Stay**, as well as three multi-step macro-actions: **Move-to-small-box(i)** and **Move-to-big-box(i)** navigate the agent to the red spot below the corresponding box and terminate with agent facing the box; **Push** operates the agent to keep moving forward until arriving the world's boundary, touching the big box along or pushing a small box to the goal. Each agent can only capture the status (*empty*, *teammate*, *boundary*,

*small or big box*) of the cell in front of it as one macro-observation. When any box is pushed to the goal, the team receives a terminal reward ($+300$ for the big box and $+20$ for each small box). A penalty $-10$ is issued when any agent hits the boundary or pushes the big box on its own.

**Overcooked** (Fig. 1b-d). Two agents must learn to co-operatively prepare a lettuce-tomato salad and deliver it to the 'star' cell as soon as possible. The challenge is that the recipe of making a lettuce-tomato salad (Fig. 2) is unknown to agents. Agents have to learn the correct procedure in terms of picking up raw vegetables, chop-

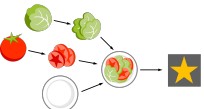

Figure 2: Lettuce-tomato salad recipe.

ping, and merging in a plate before delivering. With primitive-actions (*up*, *down*, *left*, *right*, and *stay*), agents can move around and achieve picking, placing, chopping and delivering by moving against the corresponding cell. The macro-action set consists of: a) five one-step macro-actions that are same as the primitive ones; b) **Chop**, takes three timesteps to cut a raw vegetable into pieces; c) long-term navigation macro-actions: **Get-Lettuce**, **Get-Tomato**, **Get-Plate-1/2**, **Go-Cut-Board-1/2** and **Deliver**, which move an agent to the location of the corresponding object with various possible terminal effects (e.g., holding a vegetable in hand or placing a chopped vegetable in a plate); d) **Go-Counter**, navigates an agent to one of the middle counter cells as well as placing or picking an object there. Full details of macro-actions are listed in Appendix A.3.2. Agents only observe the *positions* and *status* of the entities within a $5 \times 5$ egocentric view field. Reward mechanism involves: $+10$ for chopping a vegetable into pieces, $+200$ terminal reward for delivering a lettuce-tomato salad, $-5$ for delivering any wrong entity that is then reset to the original position, and $-0.1$ for every timestep.

**Warehouse Tool Delivery** (Fig. 1e-g). In each workshop (e.g., W-0), a human is working on an assembly task (involving 4 sub-tasks that each takes a number of timesteps to be finished) and requires three particular tools for future sub-tasks to continue. A arm robot (grey) with the duty of finding tools for each human on the table (brown) and passing them to mobile robots (green, blue and yellow) who are responsible for delivering tools to humans. Note that, the correct tools needed by each human are unknown to robots, which has to be learned during training in order to perform timely delivery without letting humans wait there. We make the original problem more challenging by: adding one more human into the domain (Fig. 1e); increasing the number of both agents and humans to further examine the scalability of our methods and the effectiveness of Mac-IAICC on handling more intricate asynchronous terminations over agents (Fig. 1f); and including one faster human (orange) to check if agents can learn a priority for assisting him (Fig. 1g).

Mobile robots move around in a certain speed in the continuous space by running one of the following macro-actions: **Go-W(i)**, moves to the waypoint (red) at a workshop; **Go-TR**, goes to the waypoint at the right side of the tool room; and **Get-Tool**, navigates to a pre-allocated waypoint besides the arm robot and waits there until either receiving a tool or 10 timesteps have passed. Applicable macro-actions for the arm robot involves: **Search-Tool(i)** lasts 6 timesteps to find the tool $i$ and place it in a staging area (containing at most two tools), otherwise freezes the robot for the same amount of time when the area is fully occupied; **Pass-to-M(i)** costs 4 timesteps to pass a tool to a mobile robot from the staging area in a first-in-first-out order; and **Wait-M**, takes 1 timestep to wait for mobile robots. Arm robot only observes the *type* of each tool in the staging area and *which mobile robot* is waiting beside. Mobile robot always detects its *position* and the *type* of each tool carried by itself, while observes the *number* of tools in the staging area or the *sub-task* a human is working on only when locating at the tool room or the workshop respectively. The team receives: $+100$ for sending a correct tool to a human in time, $-20$ for a delayed delivery, $-10$ for the arm robot running **Pass-to-M(i)** without mobile robot $i$ being next to it, and $-1$ every timestep.

## 4.2 EXPERIMENTAL IMPLEMENTATION

All methods apply the same neural network architecture for both actor and critic, which consists of two fully connected (FC) layers with Leaky-Relu activation function, one GRU layer (Cho et al., 2014) and one more FC layer followed by an output layer. In all methods, decentralized actors and decentralized critics have 32 units on FC and GRU layers, while the centralized have 64 units on the GRU layer due to dealing with larger joint macro-observation and macro-action spaces. Exploration is deployed by applying a linear decaying $\epsilon$-soft policy (Foerster et al., 2018). Hyper-parameter tuning uses grid search over a wide range of candidates (refer to Appendix A.5). The performance metric of one training trial is a mean discounted return measured by periodically (every 100 episodes)

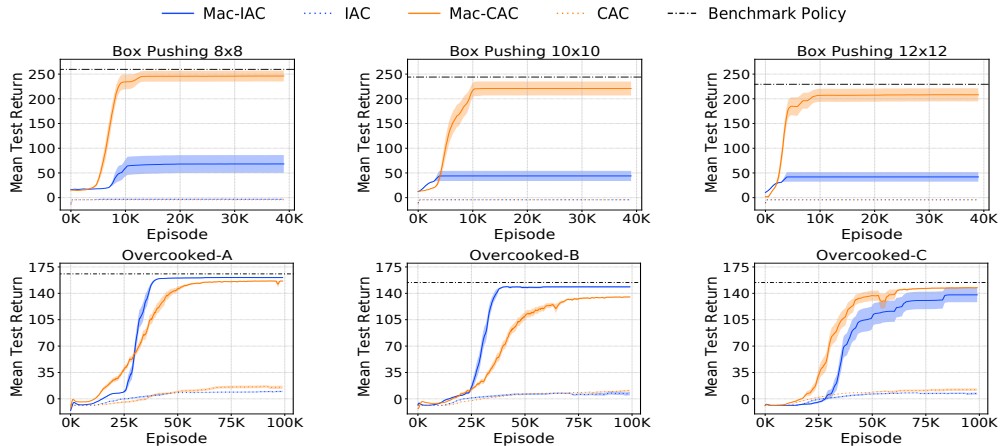

Figure 3: Decentralized learning and centralized learning with macro-actions vs primitive-actions.

evaluating the learned policies over 10 testing episodes. We plot the averaged performance of each method over 20 independent trials with one standard error and smooth the curves over 10 neighbors. The return of a benchmark policy created by using prior domain knowledge is also involved.

## 5 RESULTS AND DISCUSSIONS

**Advantages of learning with macro-actions**. We first present a comparison of our macro-action-based actor-critic methods against the primitive-action-based in fully decentralized and fully centralized cases. We consider various grid world sizes of Box Pushing domain (top row in Fig. 3, where the benchmark policy's return is the optima) and three Overcooked scenarios (bottom row in Fig. 3). The results show the significant outperformance of using macro-actions over primitive-actions. More concretely, in Box Pushing domain, reasoning about primitive movements at every timestep with such a limited observation setup makes this problem intractable to agents so that they cannot learn any good behaviors other than to keep moving around. The necessary cooperation in this task is not required in the low-level navigation but at the high-level (e.g., go to the big box and push) so that, with the macro-actions, Mac-CAC reaches near-optimal performance enabling agents to push the big box together. Unlike the centralized critic conditioning on joint information, even in the macro-action case, it is hard for each agent's decentralized critic to correctly measure the responsibility for a penalty caused by teammate pushing the big box alone. Mac-IAC thus converges to a local-optima of pushing two small boxes in order to avoid getting the penalty.

In Overcooked domain, an efficient solution requires agents to asynchronously work on independent subtasks (e.g., one agent goes to get plate while the other agent chops vegetables), which explains why Mac-IAC can solve the task well in cases, A and B. This also indicates that using local information is enough for agents to achieve high-quality behaviors. As a result, Mac-CAC learns slower since it must figure out the redundant part of joint information in the larger joint macro-level history and action spaces and it sometimes leads to a local optimum. However, in case C, the challenge from the decentralized perspective is that agents cannot observe the status of the left side due to the limited view, and there is no immediate reward caused by the agent's action at the right side. The benefit of centralized learning emerges in this case so that Mac-CAC outperforms Mac-IAC. The primitive-action-based methods begin to learn, but perform poorly in such long-horizon tasks.

**Advantages of having individual centralized critics.** Fig. 4 shows the evaluations of our macro-action-based multi-agent actor-critic methods in all three domains, where we mainly investigate the superiority of Mac-IAICC over Naive Mac-IACC on learning decentralized policies. As each agent's observation is extremely limited in Box Pushing, we allow centralized critics in both Mac-IAICC and Naive Mac-IACC to access to the state (agents' poses and boxes' positions), but joint macro-observation-action history in another two domains. In the Box Pushing task (Fig. 4's top row), Mac-IAICC and Naive Mac-IACC both show the quality of decentralized policies. However, with the growing grid world size, Naive Mac-IACC loses its value while Mac-IAICC keeps its performance near the centralized one. This is because, from each agent's perspective, the bigger the world

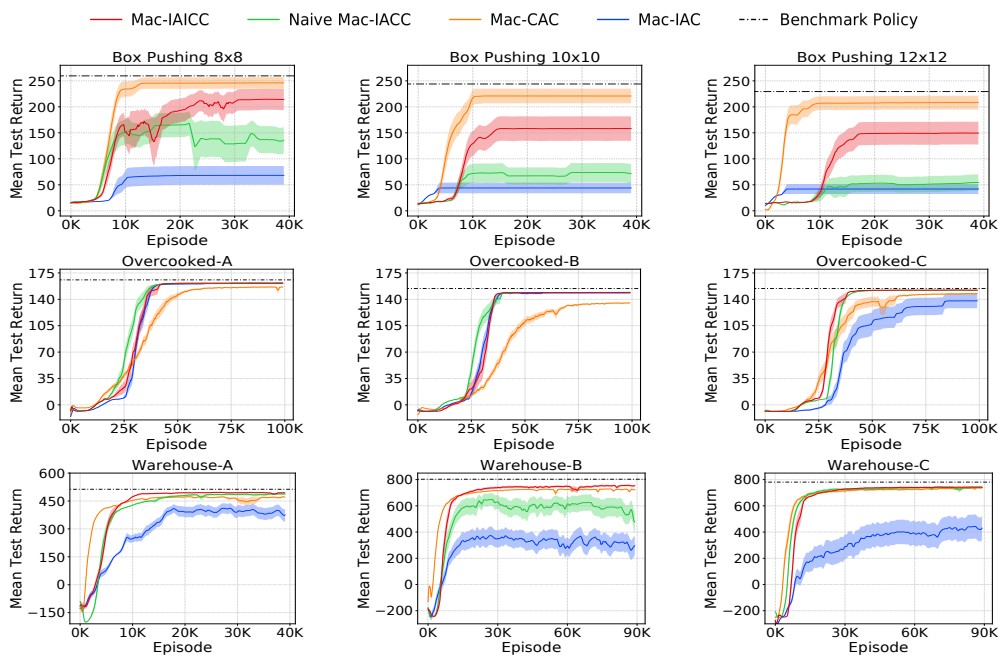

Figure 4: Comparison of asynchronous macro-action-based actor-critic methods.

size is, the more timesteps a macro-action could take, and the less accurate value estimation the critic of Naive Mac-IACC offers as it is trained depending on any agent's macro-action termination. Conversely, Mac-IAICC gives each agent a separate centralized critic trained with the reward associated with its own macro-action execution. In Overcooked-A and B (Fig. 4's mid row), as Mac-IAICC's performance is determined by the training of two agents' critics, it learns slower than Naive Mac-IACC in the early stage but both converge to the same value in the end; and in the scenario C, better decentralized policies are learned by the two CTDE-based methods. Because of the middle block in Overcooked-C, macro-action terminations from both centralized and decentralized perspectives become more frequent than that in A and B, which slows Naive Mac-IACC's learning but does not hurt the sample efficiency of Mac-IAICC. Finally, in the large warehouse domain (Fig. 4's last row), Mac-IAC performs the worst due to its natural limitations and the domain's partial observability. In particular, it is difficult for the gray robot to learn an efficient way to find correct tools purely based on local information and very delayed rewards depending on the mobile robots' behaviors. By leveraging joint information, Mac-IAICC performs the best over all three cases. Furthermore, Mac-IAICC's advantage over Naive Mac-IACC is more significant in the case B than in the case A where it converges to a higher value with much lower variance. This result confirms Mac-IAICC's scalability and effectiveness on handling more complex asynchronous executions when more agents are involved. As the difficulty for Mac-CAC discussed earlier in Overcooked, Mac-CAC also gets stuck at a local-optimum in Warehouse-A and B, and converges slightly slower than Mac-IAICC in the case C. Visualization of learned policies using Mac-IAICC are displayed in Appendix A.4.

## 6 CONCLUSION

This paper introduces the first general formulation for asynchronous multi-agent macro-action-based policy gradients under partial observability along with proposing a decentralized actor-critic method (Mac-IAC), a centralized actor-critic method (Mac-CAC), and two CTDE-based actor-critic methods (Naive Mac-IACC and Mac-IAICC). Empirically, our methods are able to learn high-quality macro-action-based policies allowing agents to perform asynchronous collaborations in large and long-horizon problems. Importantly in Mac-IAICC, the strength of allowing each agent to have an individual centralized critic associated with its own macro-acion executions clearly improves performance in many of the domains. This work provides a foundation for future macro-action-based MARL algorithm development, including other work on asynchronous execution as well as methods which also learn the macro-actions.

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

# A  APPENDIX

## A.1  MACRO-ACTION-BASED POLICY GRADIENT THEOREM

As POMDPs can always be transformed to history-based MDPs, we can directly adapt the general Bellman equation for the state values of a hierarchical policy (Sutton et al., 1999) to a macro-action-based POMDP by replacing the state $s$ with a history $h$ as follows (for keeping the notaion simple, we use $\tau$ to represent the number of timesteps taken by the corresponding macro-action $m$, and we use $h$ to represent macro-observation-action history):

$$V^{\Psi}(h) = \sum_m \Psi(m|h) Q^{\Psi}(h, m) \tag{13}$$

$$Q^{\Psi}(h, m) = r^c(h, m) + \sum_{h'} P(h'|h, m) V^{\Psi}(h') \tag{14}$$

where,

$$r^c(h, m) = \mathbb{E}_{\tau \sim \beta_m(h), s_{t_m}|h} \Big[ \sum_{t=t_m}^{t_m+\tau-1} \gamma^t r_t \Big] \tag{15}$$

$$P(h'|h, m) = P(z'|h, m) = \sum_{\tau=1}^{\infty} \gamma^{\tau} P(z', \tau|h, m) \tag{16}$$

$$= \sum_{\tau=1}^{\infty} \gamma^{\tau} P(\tau|h, m) P(z'|h, m, \tau) \tag{17}$$

$$= \sum_{\tau=1}^{\infty} \gamma^{\tau} P(\tau|h, m) P(z'|h, m, \tau) \tag{18}$$

$$= \mathbb{E}_{\tau \sim \beta_m(h)} \Big[ \gamma^{\tau} \mathbb{E}_{s|h} \big[ \mathbb{E}_{s'|s, m, \tau} [P(z'|m, s')] \big] \Big] \tag{19}$$

Next, we follow the proof of the policy gradient theorem (Sutton et al., 2000):

$$\nabla_\theta V^{\Psi_\theta}(h) = \nabla_\theta \Bigg[ \sum_m \Psi_\theta(m|h) Q^{\Psi_\theta}(h, m) \Bigg] \tag{20}$$

$$= \sum_m \Big[ \nabla_\theta \Psi_\theta(m|h) Q^{\Psi_\theta}(h, m) + \Psi_\theta(m|h) \nabla_\theta Q^{\Psi_\theta}(h, m) \Big] \tag{21}$$

$$= \sum_m \Big[ \nabla_\theta \Psi_\theta(m|h) Q^{\Psi_\theta}(h, m) + \Psi_\theta(m|h) \nabla_\theta \big( r^c(h, m) + \sum_{h'} P(h'|h, m) V^{\Psi_\theta}(h') \big) \Big] \tag{22}$$

$$= \sum_m \Big[ \nabla_\theta \Psi_\theta(m|h) Q^{\Psi_\theta}(h, m) + \Psi_\theta(m|h) \sum_{h'} P(h'|h, m) \nabla_\theta V^{\Psi_\theta}(h') \big) \Big] \tag{23}$$

$$= \sum_{\hat{h} \in H} \sum_{k=0}^{\infty} P(h \to \hat{h}, k, \Psi_\theta) \sum_m \nabla_\theta \Psi_\theta(m|\hat{h}) Q^{\Psi_\theta}(\hat{h}, m) \quad \text{(after repeated unrolling)} \tag{24}$$

Then, we can have:

$$\nabla_\theta J(\theta) = \nabla_\theta V^{\Psi_\theta}(h_0) \tag{25}$$

$$= \sum_{h \in H} \sum_{k=0}^{\infty} P(h_0 \to h, k, \Psi_\theta) \sum_m \nabla_\theta \Psi_\theta(m|h) Q^{\Psi_\theta}(h, m) \tag{26}$$

$$= \sum_h \rho^{\Psi_\theta}(h) \sum_m \nabla_\theta \Psi_\theta(m|h) Q^{\Psi_\theta}(h, m) \tag{27}$$

$$= \sum_h \rho^{\Psi_\theta}(h) \sum_m \Psi_\theta(m|h) \nabla_\theta \log \Psi_\theta(m|h) Q^{\Psi_\theta}(h, m) \tag{28}$$

$$= \mathbb{E}_{\Psi_\theta} \Big[ \nabla_\theta \log \Psi_\theta(m|h) Q^{\Psi_\theta}(h, m) \Big] \tag{29}$$

## A.2 ALGORITHM

In this section, we present the pesudo code of each proposed macro-action-based actor-critic algorithm with an example to show how the sequential experiences are squeezed for training the critic and the actor. We describe all methods in the on-policy learning manner while off-policy learning can be achieved by applying importance sampling weights and not resetting the buffer.

### A.2.1 MAC-IAC

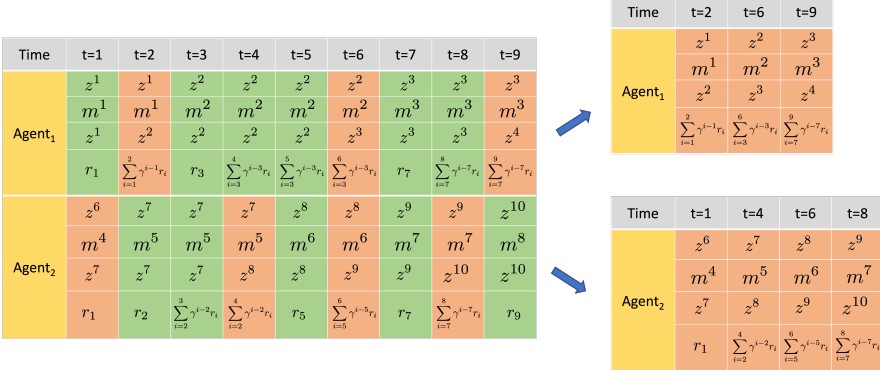

Figure 5: An example of the trajectory squeezing process in Mac-IAC. We collect each agent's high-level transition tuple at every primitive-step. Each agent is allowed to obtain a new macro-observation if and only if the current macro-action terminates, otherwise, the next macro-observation is set as same as the previous one. Each agent separately squeezes its sequential experiences by picking out the transitions when its macro-action terminates (red cells). Each agent independently train the critic and the policy using the squeezed trajectory.

---

**Algorithm 1** Mac-IAC

1: Initialize a decentralized policy network for each agent $i$: $\Psi_{\theta_i}$
2: Initialize decentralized critic networks for each agent $i$: $V_{\mathbf{w}_i}^{\Psi_{\theta_i}}, V_{\mathbf{w}_i^-}^{\Psi_{\theta_i}}$
3: Initialize a buffer $\mathcal{D}$
4: **for** *episode* = 1 to $M$ **do**
5:      $t = 0$
6:      Reset env
7:      **while** not reaching a terminal state **and** $t < \mathbb{H}$ **do**
8:          $t \leftarrow t + 1$
9:          **for** each agent $i$ **do**
10:             **if** the macro-action $m_i$ is terminated **then**
11:               $m_i \sim \Psi_{\theta_i}(\cdot \mid h_i; \epsilon)$
12:             **else**
13:               Continue running current macro-action $m_i$
14:          **for** each agent $i$ **do**
15:             Get cumulative reward $r_i^c$, next macro-observation $z_i'$
16:             Collect $\langle z_i, m_i, z_i', r_i^c \rangle$ into the buffer $\mathcal{D}$
17:      **if** *episode* mod $I_{\text{train}} = 0$ **then**
18:          **for** each agent $i$ **do**
19:             Squeeze agent $i$'s trajectories in the buffer $\mathcal{D}$
20:             Perform a gradient decent step on $L(\mathbf{w}_i) = \left(y - V_{\mathbf{w}_i}^{\Psi_{\theta_i}}(h_i)\right)_{\mathcal{D}}^2$, where $y = r_i^c + \gamma^{\tau_{m_i}} V_{\mathbf{w}_i^-}^{\Psi_{\theta_i}}(h_i')$
21:             Perform a gradient ascent on:
22:             $\nabla_{\theta_i} J(\theta_i) = \mathbb{E}_{\vec{\Psi}_{\vec{\theta}}}\left[\nabla_{\theta_i} \log \Psi_{\theta_i}(m_i|h_i)\left(r_i^c + \gamma^{\tau_{m_i}} V_{\mathbf{w}_i^-}^{\Psi_{\theta_i}}(h_i') - V_{\mathbf{w}_i}^{\Psi_{\theta_i}}(h_i)\right)\right]$
23:             Reset buffer $\mathcal{D}$
24:      **if** *episode* mod $I_{\text{TargetUpdate}} = 0$ **then**
25:          **for** each agent $i$ **do**
26:             Update the critic target network $\mathbf{w}_i^- \leftarrow \mathbf{w}_i$

### A.2.2 MAC-CAC

Figure 6: An example of the trajectory squeezing process in Mac-CAC. Joint sequential experiences are squeezed by picking out joint transition tuples when the joint macro-action terminates, in that, *any* agent's macro-action termination (marked in red) ends the joint macro-action at the timestep. For example, at $t = 1$, agents execute a joint macro-action $\vec{m} = \langle m^1, m^4 \rangle$ for one timestep; at $t = 2$, the joint macro-action becomes $\langle m^1, m^5 \rangle$ as Agent$_2$ finished $m^4$ at last step and chooses a new macro-action $m^5$; Agent$_1$ finished its macro-action $m_1$ at $t = 2$ and selects a new macro-action $m^2$ at $t = 3$ so that the joint macro-action switches to $\langle m^2, m^5 \rangle$ which keeps running until the 4th timestep. Therefore, the first two joint macro-actions have two single-step reward respectively, and reward of joint macro-action $\langle m^2, m^5 \rangle$ is an accumulative reward over two consecutive timesteps.

---

**Algorithm 2** Mac-CAC

---

1: Initialize a centralized policy network: $\Psi_\theta$
2: Initialize centralized critic networks: $V_{\mathbf{w}}^{\Psi_\theta}, V_{\mathbf{w}^-}^{\Psi_\theta}$
3: Initialize a centralized buffer $\mathcal{D} \leftarrow$ Mac-JERTs,
4: **for** *episode* = 1 to $M$ **do**
5:     $t = 0$
6:     Reset env
7:     **while** not reaching a terminal state **and** $t < \mathbb{H}$ **do**
8:         $t \leftarrow t + 1$
9:         **if** the joint macro-action $\vec{m}$ is terminated **then**
10:             $\vec{m} \sim \Psi_\theta(\cdot \mid \vec{h}, \vec{m}^{\text{undone}}; \epsilon)$
11:         **else**
12:             Continue running current joint macro-action $\vec{m}$
13:         Get a joint cumulative reward $\vec{r}^c$, next joint macro-observation $\vec{z}'$
14:         Collect $\langle \vec{z}, \vec{m}, \vec{z}', \vec{r}^c \rangle$ into the buffer $\mathcal{D}$
15:     **if** *episode* mod $I_{\text{train}} = 0$ **then**
16:         Squeeze joint macro-level trajectories in the buffer $\mathcal{D}$ according to joint macro-action terminations
17:         Perform a gradient decent step on $L(\mathbf{w}) = \left( y - V_{\mathbf{w}}^{\Psi_\theta}(\vec{h}) \right)_{\mathcal{D}}^2$, where $y = \vec{r}^c + \gamma^{\vec{\tau}_{\vec{m}}} V_{\mathbf{w}^-}^{\Psi_\theta}(\vec{h}')$
18:         Perform a gradient ascent on $\nabla_\theta J(\theta) = \mathbb{E}_{\Psi_\theta} \left[ \nabla_\theta \log \Psi_\theta(\vec{m} \mid \vec{h}) \left( \vec{r}^c + \gamma^{\vec{\tau}_{\vec{m}}} V_{\mathbf{w}^-}^{\Psi_\theta}(\vec{h}') - V_{\mathbf{w}}^{\Psi_\theta}(\vec{h}) \right) \right]$
19:         Reset buffer $\mathcal{D}$
20:     **if** *episode* mod $I_{\text{TargetUpdate}} = 0$ **then**
21:         Update the critic target network $\mathbf{w}^- \leftarrow \mathbf{w}$

---

where, $\vec{m}^{\text{undone}}$ is the sub-joint-macro-action over the agents who have not terminated their macro-actions and will continue running.

### A.2.3 NAIVE MAC-IACC

In the pseudo code of Naive Mac-IACC presented below, we assume the accessible centralized information $\mathbf{x}$ is joint macro-observation-action history in the centralized critic.

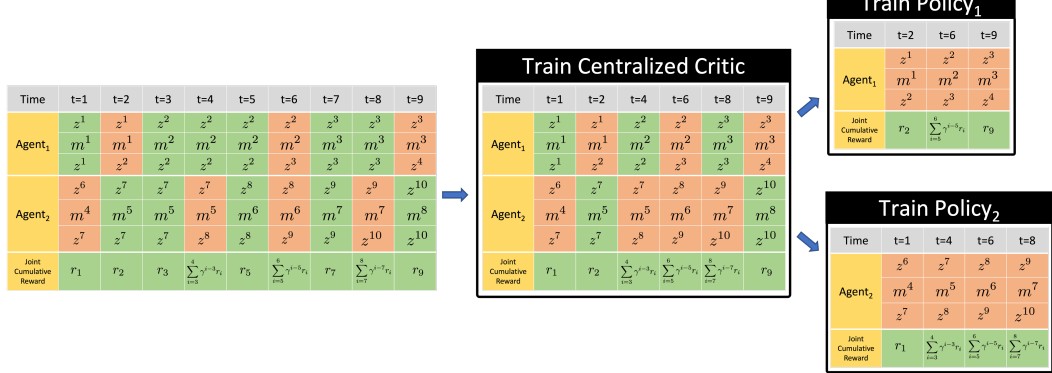

Figure 7: An example of the trajectory squeezing process in Navie Mac-IACC. The joint trajectory is first squeezed depending on joint macro-action termination for training the centralized critic (line 18-19 in Algorithm 3). Then, the trajectory is further squeezed for each agent depending on each agent's own macro-action termination for training the decentralized policy (line 20-23 in Algorithm 3.

---

**Algorithm 3** Naive Mac-IACC
___

1:  Initialize a decentralized policy network for each agent $i$: $\Psi_{\theta_i}$
2:  Initialize centralized critic networks: $V_{\mathbf{w}}^{\vec{\Psi}_{\vec{\theta}}}, V_{\mathbf{w}^-}^{\vec{\Psi}_{\vec{\theta}}}$
3:  Initialize a decentralized buffer $\mathcal{D} \leftarrow$ Mac-JERTs,
4:  **for** *episode* = 1 to $M$ **do**
5:      $t = 0$
6:      Reset env
7:      **while** not reaching a terminal state **and** $t < \mathbb{H}$ **do**
8:          $t \leftarrow t + 1$
9:          **for** each agent $i$ **do**
10:             **if** the macro-action $m_i$ is terminated **then**
11:                 $m_i \sim \Psi_{\theta_i}(\cdot \mid h_i; \epsilon)$
12:             **else**
13:                 Continue running current macro-action $m_i$
14:         Get a reward $\vec{r}^{\,c}$ accumulated based on current joint macro-action termination
15:         Get next joint macro-observations $\vec{z}'$
16:         Collect $\langle \vec{z}, \vec{m}, \vec{z}', \vec{r}^{\,c} \rangle$ into the buffer $\mathcal{D}$
17:      **if** *episode* mod $I_{\text{train}} = 0$ **then**
18:         Squeeze joint macro-level trajectories in the buffer $\mathcal{D}$ according to joint macro-action terminations
19:         Perform a gradient decent step on $L(\mathbf{w}) = \left(y - V_{\mathbf{w}}^{\vec{\Psi}_{\vec{\theta}}}(\vec{h})\right)_{\mathcal{D}}^2$, where $y = \vec{r}^{\,c} + \gamma^{\vec{\tau}_{\vec{m}}} V_{\mathbf{w}^-}^{\vec{\Psi}_{\vec{\theta}}}(\vec{h}')$
20:         **for** each agent $i$ **do**
21:             Squeeze agent $i$'s trajectories in the buffer $\mathcal{D}$ according to its own macro-action terminations
22:             Perform a gradient ascent on:
23:                 $\nabla_{\theta_i} J(\theta_i) = \mathbb{E}_{\vec{\Psi}_{\vec{\theta}}}\left[\nabla_{\theta_i} \log \Psi_{\theta_i}(m_i|h_i)\left(\vec{r}^{\,c} + \gamma^{\vec{\tau}_{\vec{m}}} V_{\mathbf{w}^-}^{\vec{\Psi}_{\vec{\theta}}}(\vec{h}') - V_{\mathbf{w}}^{\vec{\Psi}_{\vec{\theta}}}(\vec{h})\right)\right]$
24:         Reset buffer $\mathcal{D}$
25:      **if** *episode* mod $I_{\text{TargetUpdate}} = 0$ **then**
26:         Update the critic target network $\mathbf{w}^- \leftarrow \mathbf{w}$

### A.2.4 MAC-IAICC

In the pseudo code of Mac-IAICC presented below, we assume the accessible centralized information $\mathbf{x}$ is joint macro-observation-action history in the centralized critic.

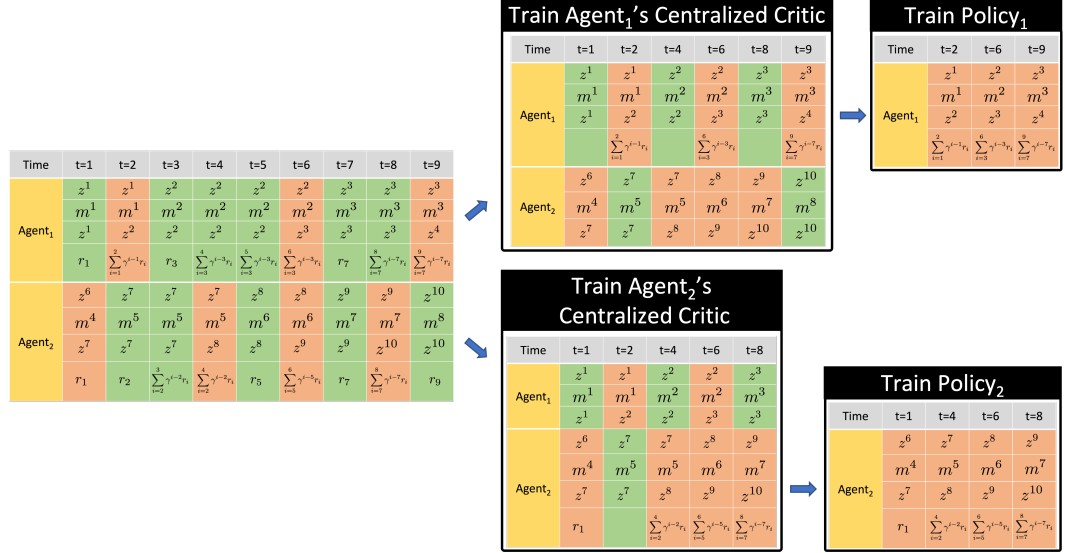

Figure 8: An example of the trajectory squeezing process in Mac-IAICC: each agent learns an individual centralized critic for the decentralized policy optimization. In order to achieve a better use of centralized information, the recurrent layer in each critic's neural network should receive all the valid joint macro-observation-action information (when $any$ agent terminates its macro-action (line 20-22) and obtain a new joint macro-observation). However, the critic's TD updates and the policy's updates still rely on each agent's individual macro-action termination and the accumulative reward at the corresponding timestep (line 23-26). Hence, the trajectory squeezing process for training each critic still depends on joint-macro-action termination but only retaining the accumulative rewards w.r.t. the corresponding agent's macro-action termination for computing the TD loss (the middle part in the above picture). Then, each agent's trajectory is further squeezed depending on its macro-action termination to update the decentralized policy.

---

**Algorithm 4** Mac-IAICC

---

1: Initialize a decentralized policy network for each agent $i$: $\Psi_{\theta_i}$
2: Initialize centralized critic networks for each agent $i$: $V_{\mathbf{w}_i}^{\vec{\Psi}_{\vec{\theta}}}$, $V_{\mathbf{w}_i^-}^{\vec{\Psi}_{\vec{\theta}}}$
3: Initialize a decentralized buffer $\mathcal{D}$
4: **for** *episode* = 1 to $M$ **do**
5:     $t = 0$
6:     Reset env
7:     **while** not reaching a terminal state **and** $t < \mathbb{H}$ **do**
8:         $t \leftarrow t + 1$
9:         **for** each agent $i$ **do**
10:             **if** the macro-action $m_i$ is terminated **then**
11:                 $m_i \sim \Psi_{\theta_i}(\cdot \mid h_i; \epsilon)$
12:             **else**
13:                 Continue running current macro-action $m_i$
14:         **for** each agent $i$ **do**
15:             Get a reward $r_i^c$ accumulated based on agent $i$'s macro-action termination
16:         Get next joint macro-observations $\vec{z}'$
17:         Collect $\langle \vec{z}, \vec{m}, \vec{z}', \{r_1^c, \ldots, r_n^c\} \rangle$ into the buffer $\mathcal{D}$
18:     **if** *episode* mod $I_{\text{train}} = 0$ **then**
19:         **for** each agent $i$ **do**
20:             Squeeze trajectories in the buffer $\mathcal{D}$ according to joint macro-action terminations
21:             Compute the TD-error of each timestep in the squeezed experiences:
22:             $L(\mathbf{w}_i) = \left(y - V_{\mathbf{w}_i}^{\vec{\Psi}_{\vec{\theta}}}(\vec{h})\right)_{\mathcal{D}}^2$, where $y = r_i^c + \gamma^{\tau_{m_i}} V_{\mathbf{w}_i^-}^{\vec{\Psi}_{\vec{\theta}}}(\vec{h}')$
23:             Perform a gradient descent only over the TD-errors when agent $i$'s macro-action is terminated
24:             Squeeze agent $i$'s trajectories in the buffer $\mathcal{D}$ according to its own macro-action terminations
25:             Perform a gradient ascent on:
26:             $\nabla_{\theta_i} J(\theta_i) = \mathbb{E}_{\vec{\Psi}_{\vec{\theta}}}\left[\nabla_{\theta_i} \log \Psi_{\theta_i}(m_i|h_i)\left(r_i^c + \gamma^{\tau_{m_i}} V_{\mathbf{w}_i^-}^{\vec{\Psi}_{\vec{\theta}}}(\vec{h}') - V_{\mathbf{w}_i}^{\vec{\Psi}_{\vec{\theta}}}(\vec{h})\right)\right]$
27:         Reset buffer $\mathcal{D}$
28:     **if** *episode* mod $I_{\text{TargetUpdate}} = 0$ **then**
29:         **for** each agent $i$ **do**
30:             Update the critic target network $\mathbf{w}_i^- \leftarrow \mathbf{w}_i$

---

### A.3 DOMAIN DESCRIPTION AND RESULTS

### A.3.1 BOX PUSHING

**Domain Setup.**

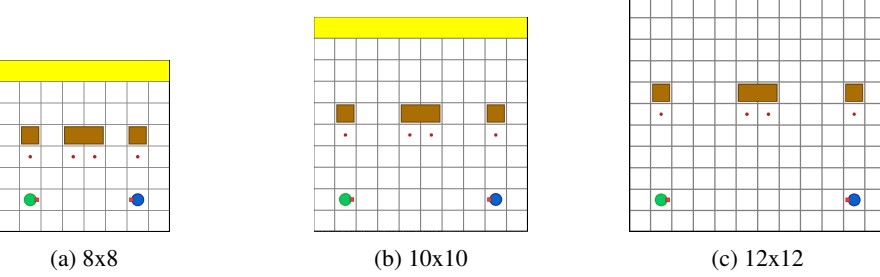

(a) 8x8     (b) 10x10     (c) 12x12

Figure 9: Experimental environments.

**Goal**. The objective of the two agents is to learn collaboratively push the middle big box to the goal area at the top rather than pushing a small box on each own.

**State**. The global state information consists of the position and orientation of each agent and each box's position in a grid world.

**Primitive-Action Space**. *move forward*, *turn-left*, *turn-right* and *stay*.

**Macro-Action Space**.

• One-step macro-actions: ***Turn-left***, ***Turn-right***, and ***Stay***.

• Multi-step macro-actions: ***Move-to-small-box(i)*** that navigates the agent to the red spot below the corresponding small box and terminate with agent facing the box; ***Move-to-big-box(i)*** that navigates the agent to a red spot below the big box and terminate with agent facing the big box; ***Push*** that operates the agent to keep moving forward and terminate while arriving the world's boundary, touching the big box along or pushing a small box to the goal.

**Observation Space**. In both the primitive-observation and macro-observation, each agent is only allowed to capture one of five states of the cell in front of it: *empty*, *teammate*, *boundary*, *small box*, *big box*.

**Dynamics**. The transition in this task is deterministic. Boxes can only be moved towards the north when the agent faces the box and moves forward. The small box can be moved by a single agent while the big box require two agents to move it together.

**Rewards**. The team receives $+300$ for pushing big box to the goal area and $+20$ for pushing a small box to the goal area. A penalty $-10$ is issued when any agent hits the boundary or pushes the big box on its own.

**Episode Termination**. Each episode terminates when any box is pushed to the goal area, or when 100 timesteps has elapsed.

**Results**

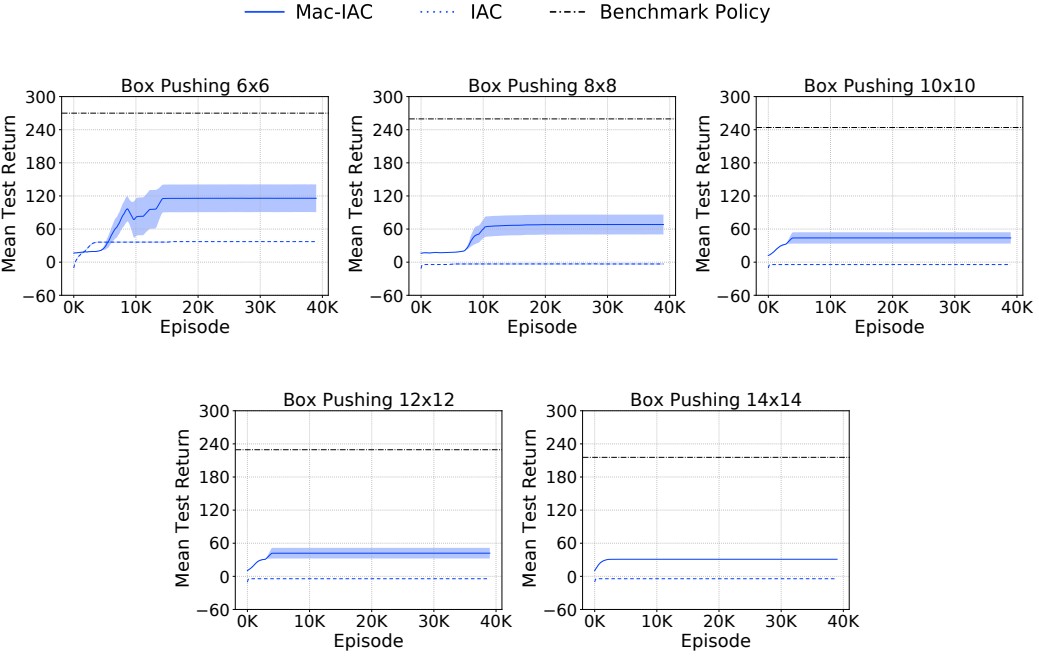

Figure 10: Decentralized learning with macro-actions vs primitive-actions in Box Pushing domain.

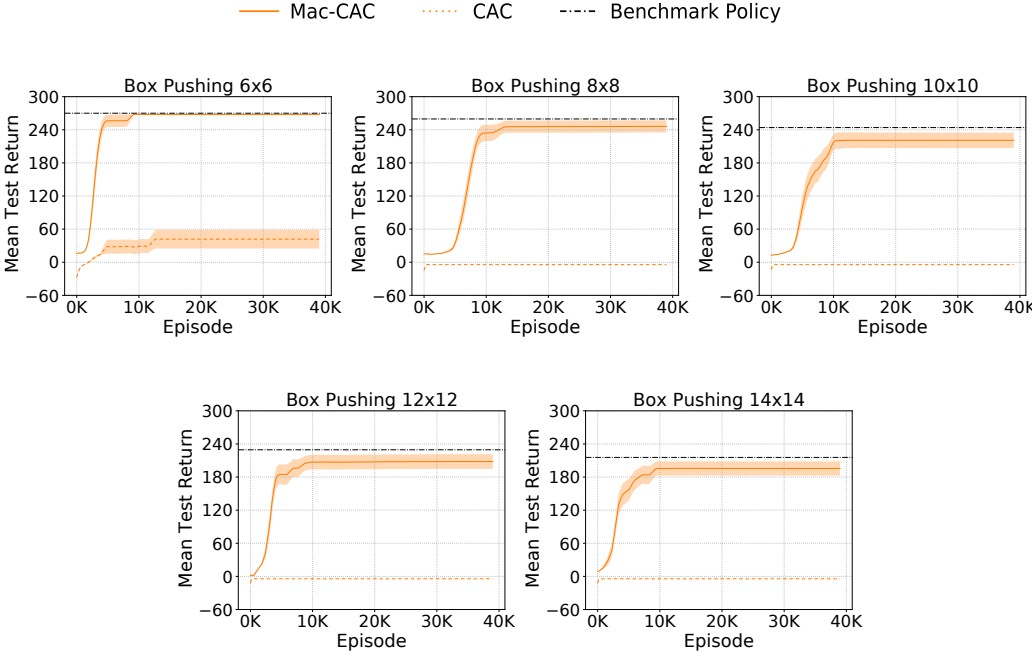

Figure 11: Centralized learning with macro-actions vs primitive-actions in Box Pushing domain.

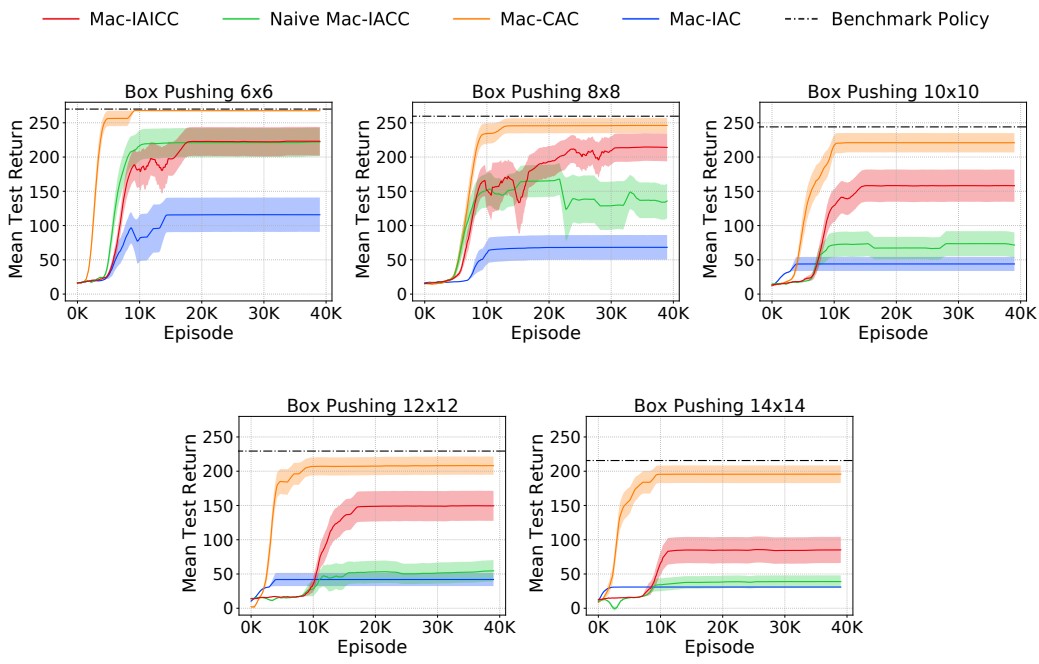

Figure 12: Comparison of macro-action-based multi-agent actor-critic methods in Box Pushing.

### A.3.2 OVERCOOKED

**Domain Setup**

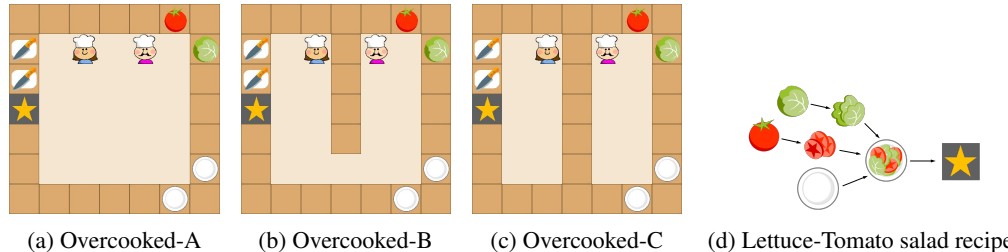

(a) Overcooked-A     (b) Overcooked-B     (c) Overcooked-C     (d) Lettuce-Tomato salad recipe

Figure 13: Experimental environments.

**Goal**. Two agents need to learn cooperating with each other to prepare a Lettuce-Tomato salad and deliver it to the 'star' counter cell as soon as possible. The challenge is that the recipe of making a lettuce-tomato salad (Fig. 2) is unknown to agents. Agents have to learn the correct procedure in terms of picking up raw vegetables, chopping, and merging in a plate before delivering.

**State Space**. The environment is a 7×7 grid world involving two agents, one tomato, one lettuce, two plates, two cutting boards and on delivery cell. The global state information consists of the location of each agent and above items, and the status of each vegetable: chopped, unchopped, or the progress under chopping.

**Primitive-Action Space**. Each agent has five primitive-actions: *up*, *down*, *left*, *right* and *stay*. Agents can move around and achieve picking, placing, chopping and delivering by moving against the corresponding counter cell (e.g. standing at the cell next to the lettuce and executing *right* to pick it up.

**Macro-Action Space**.

• One-step macro-actions:

*Stay*, stay at the current position and terminate.

*Up*, *Down*, *Left* and *Right*, each moves the agent to the corresponding nearby cell and terminate.

• Multi-step macro-actions:

*Get-Tomato* and *Get-Lettuce*, navigate the agent to the latest observed position of the vegetable. If the vegetable is there, pick it up, otherwise the agent moves to the initial position of the vegetable, and picks it up if the vegetable is there, otherwise terminates the macro-action.

Termination conditions:

Case 1: The agent is next to the chopped/unchopped food and picks it up.
Case 2: The agent observes that the food is being held by the other agent or itself.
Case 3: The agent is next to the unchopped food holding a plate or other food.
Case 4: The agent is next to the other agent and this agent is next to the food.
Case 5: The agent is next to the cutting board counter and the unchopped food is on the counter.
Case 6: The agent does not find the tomato/lettuce in the latest position it observed it. And the agent goes to the initial position of the food and does not find it, either.
Case 7: Two agents attempt to enter the same cell. One agent stays and terminates according to a predefined priority.

*Get-Plate-1/2*, navigates the agent to the latest observed position of the plate. If the plate is there, pick it up, otherwise the agent moves to the initial position of the plate, and picks it up if the plate is there, otherwise terminates the macro-action.

Termination conditions:

Case 1: The agent is next to the plate and picks it up.
Case 2: The agent observes that the plate is being held by the other agent or itself.
Case 3: The agent is next to the plate but holding another plate or unchopped food.
Case 4: The agent is next to the other agent and this agent is next to the plate.
Case 5: The agent does not find the plate in the latest position it observed it. And the agent goes to the initial position of the plate and does not find it, either.
Case 6: Two agents attempt to enter the same cell. One agent stays and terminates according to a predefined priority.

*Go-Cut-Board-1/2*, navigates the agent to the corresponding cutting board.

Termination conditions:

Case 1: The agent is next to the cutting board counter without holding anything.
Case 2: The cutting board is empty. The agent is next to the cutting board while holding plate or food, and then put it on the cutting board.
Case 3: There is a plate or food on the cutting board. The agent stops at the cell next to the cutting board.
Case 4: The cutting board is being used by teammate, the agent stops at the cell next to the teammate.
Case 5: Two agents attempt to enter the same cell. One agent stays and terminates according to a predefined priority.

*Chop*, chops the raw vegetable into pieces, which takes three timesteps.

Termination conditions:

Case 1: The vegetable on the cutting board has been chopped into pieces.
Case 2: Immediately terminates when the agent is not next to a cutting board.
Case 3: Immediately terminates when there is no unchopped food on the cutting board.
Case 4: The agent holds something.

***Deliver***, navigates the agent to the 'star' counter cell for delivering by putting down the in-hand object.

Termination conditions:

Case 1: The agent is next to the delivery counter without holding anything.
Case 2: The agent is next to the delivery counter holding a object and put down the object on the delivery counter cell.
Case 3: Teammate is standing in front of the delivery counter cell, the agent terminates at the cell next to teammate.
Case 4: Two agents attempt to enter the same cell. One agent stays and terminates according to a predefined priority.

***Go-Counter***, navigates the agent to one empty counter cell in the middle of the map. The priority of the targeting counter cell is from middle to the sides. This macro-action is only available in map B and C.

Termination conditions:

Case 1: The agent is next to the counter cell without holding anything.
Case 2: The agent is next to the counter cell that is not empty.
Case 3: The agent is next to the counter cell and puts in-hand object on it.
Case 4: Teammate is in front of the target counter cell, the agent then stops next to the teammate.
Case 5: Two agents attempt to enter the same cell. One agent stays and terminates according to a predefined priority.

**Observation Space**: The macro-observation space for each agent is the same as the primitive observation space. Agents are only allowed to observe the *positions* and *status* of the entities within a $5 \times 5$ egocentric view field. The initial position of all the items are known to the agents.

**Dynamics**: The transition in this task is deterministic. If an agent delivers any wrong item, the item will be reset to its initial position. From the low-level perspective, to chop a vegetable into pieces on a cutting board, the agent needs to stand next to the cutting board and executes *left* three times. Only the chopped food can be put on a plate.

**Reward**: $+10$ for chopping a vegetable, $+200$ terminal reward for delivering a lettuce-tomato salad, $-5$ for delivering any wrong entity, and $-0.1$ for every timestep.

**Episode Termination**: Each episode terminates either when agents successfully deliver a lettuce-tomato salad correct dish or reaching the maximal timesteps, 200.

**Results**

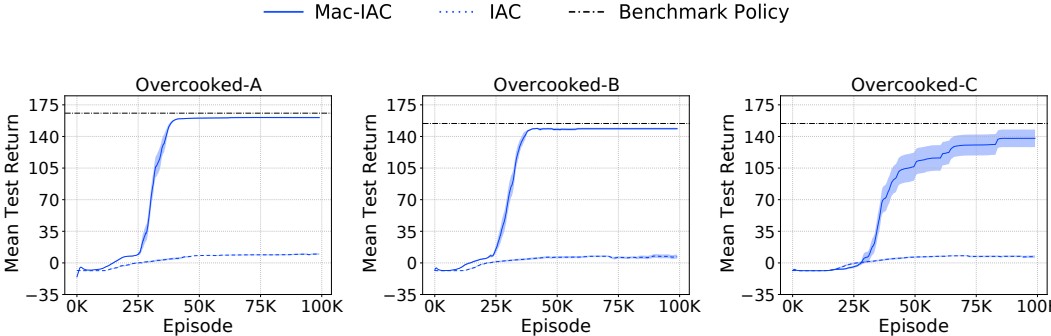

Figure 14: Decentralized learning with macro-actions vs primitive-actions in Overcooked domain.

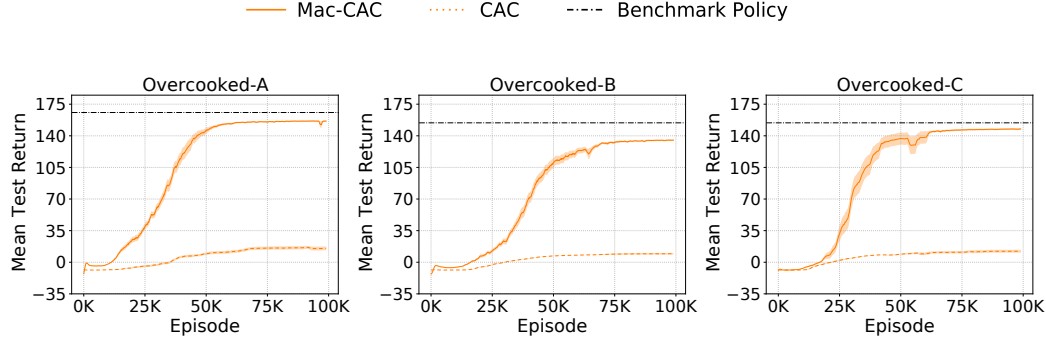

Figure 15: Centralized learning with macro-actions vs primitive-actions in Overcooked domain.

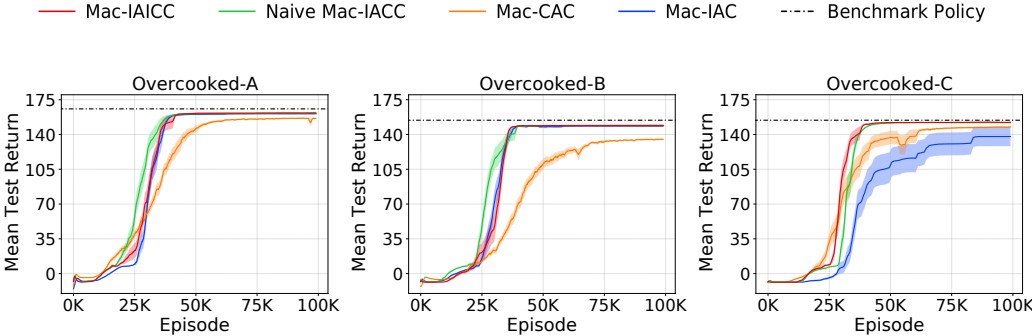

Figure 16: Comparison of macro-action-based multi-agent actor-critic methods in Overcooked.

### A.3.3 Warehouse Tool Delivery

**Domain Setup**

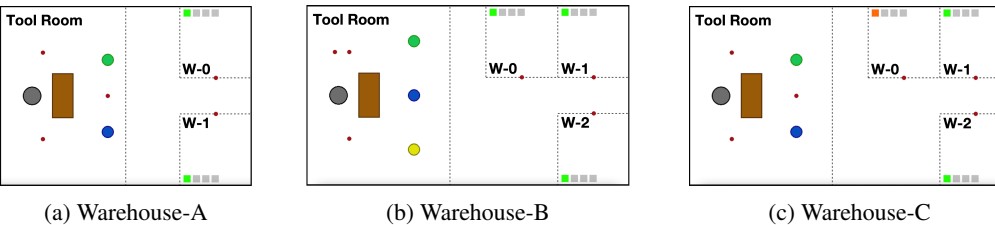

|  |  |  |
|---|---|---|
| (a) Warehouse-A | (b) Warehouse-B | (c) Warehouse-C |

Figure 17: Experimental environments.

In this Warehouse Tool Delivery domain, we consider three different scenarios shown in Fig. 17. We make the original problem (Xiao et al., 2019) (only one human worker involved) more challenging by: 1) adding one more human into the environment (Fig. 17a); 2) increasing the number of agents and having three humans in the environment to further examine the scalability of our methods and the effectiveness of Mac-IAICC on handling more noisy asynchronous terminations over agents (Fig. 17a); 3) having one faster human involved to check whether the agents can learn a priority for delivering tool to him or not.

**Goal.** Under all scenarios, in each workshop, a human is working on an assembly task involving 4 subtasks to be finished (each subtask takes amount of primitive timesteps) and requires three particular tools for each future subtask to continue. Humans either work in the same speed (Fig. 17a-b) or have one of them working faster (the orange one in Fig.17c). The robot team includes a arm robot (grey) with the duty of finding tools for each human on the table (brown) and passing them to mobile robots (green, blue and yellow) who are responsible for delivering tools to the humans. The objective of the robots is to assist the humans to finish their assembly tasks as soon as possible by finding the correct tools in a efficient order. To make this problem more challenging, the correct tools that each human needs are unknown to the robots, which has to be learned during training in order to perform timely delivery without letting any human wait over there.

**State**. The environment is either a $5 \times 7$ (Fig. 17a) or a $5 \times 9$ (Fig. 17b-c) continuous space. A global state consists of the 2D position of each mobile robots, the execution status of the arm robot's current macro-action (e.g how munch steps are left for completing the macro-action, but in real-world, this should be the angle and speed of each arm's joint), the subtask each human is working with a percentage indicating the progress of the subtask, and the position of each tools (either on the brown table or carried by a mobile robot). Note that, there are enough tools for each human such that the number of each type of tool exactly matches with the number of humans in the environment. The initial state of every episode is always same as shown in Fig. 17, where humans always start from the first step.

**Macro-Action Space**.

The available macro-actions for each mobile robot include:

• *Go-W(i)*, navigates to the red waypoint at the corresponding workshop;

• *Go-TR*, navigates to the red waypoint at the right side of the tool room;

• *Get-Tool*, navigates to a pre-allocated waypoint besides the arm robot and waits over there until either 10 timesteps have passed or receiving a tool from the gray robot.

The available macro-actions for the arm robot include:

• *Search-Tool(i)*, takes 6 timesteps to find tool $i$ and place it in a staging area (containing at most two tools) when the area is not fully occupied, otherwise freezes the robot for the same amount of time;

• *Pass-to-M(i)*, takes 4 timesteps to pass a tool to a mobile robot from the staging area in the order of first-in-first-out;

• *Wait-M*, takes 1 timestep to wait for mobile robots coming.

**Macro-Observation Space**.

The arm robot's macro-observation include the information about *the type* of each tool in the staging area and *which mobile robot* is waiting beside.

Each mobile robot always observes its own *position* and *the type* of each tool carried by itself, while observes *the number* of tools in the staging area or *the subtask* a human working on only when locating at the tool room or the workshop respectively.

**Dynamics**. Transitions are deterministic. Each mobile robot moves in a speed 0.8 and is only allowed to receive tools from the arm robot rather than from humans. Each human is only allowed to possess the tool for the next subtask from a mobile robot when the robot locates at the corresponding workshop and carries the exact tool. In the Warehouse-A, each human takes 18 timesteps to finish each subtask; in the Warehouse-B, each human takes 40 timesteps to finish each subtask; and in the Warehouse-C, each subtask takes the faster human 30 timesteps while taking the slower human 40 timesteps. Human cannot start the next subtask without obtaining the correct tool.

**Rewards**. The team receives a $+100$ reward when a correct tool is delivered to a human in time while getting an extra $-20$ penalty for a delayed delivery such that the human has paused over there. A $-10$ reward occurs when the grey robot does ***Pass-to-M(i)*** but the mobile robot $i$ is not next to it, and a $-1$ reward is issued every timestep.

**Episode Termination**. Each episode terminates when all humans obtained all the correct tools for all subtasks, otherwise, the episode will run until the maximal timesteps (200 for Warehouse-A and 250 for Warehouse-B and C).

**Results**

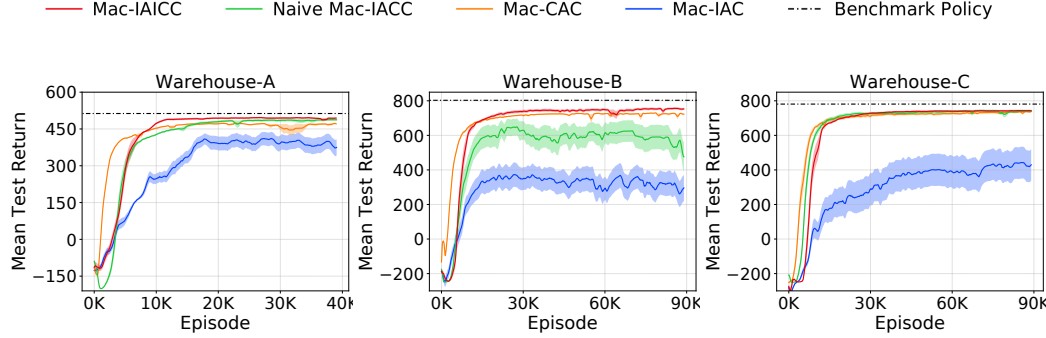

Figure 18: Comparison of macro-action-based multi-agent actor-critic methods in Warehouse Tool Delivery domain.

## A.4 BEHAVIOR VISUALIZATION

In the section, we display the decentralized behaviors learned by using Mac-IAICC under all considered domains.

### A.4.1 BOX PUSHING

We show the behaviors learned under the grid world size $14 \times 14$ in Fig. 19. Although the averaged performance of the training is not near-optimal (Fig. 12), several runs can learn the optimal behavior.

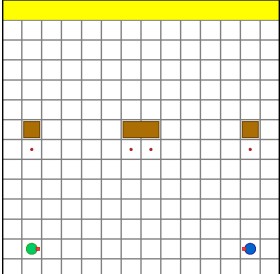 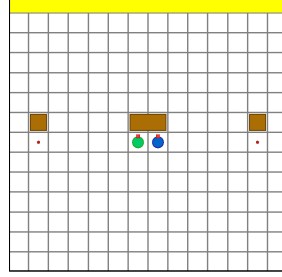 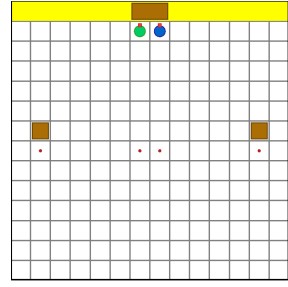

(a) Green agent executes ***Move-to-big-box(1)*** to move to the left waypoint below the big box while the blue agent runs ***Move-to-big-box(2)*** to move to the right waypoint below the big box.

(b) After completing the previous macro-actions, agents choose ***Push*** to move the big box towards the goal together.

(c) Agents finish the task by pushing the big box to the goal area.

Figure 19: Visualization of the optimal macro-action-based behaviors learned using Mac-IAICC in the Box Pushing domain under a $14 \times 14$ grid world.

### A.4.2 OVERCOOKED

**Map A.** In this map, a simple collaborative strategy is that two agents pick up the tomato and lettuce, bring them to the cutting board, chop them into pieces, and then let one agent go to get one plate, put the chopped food on the plate and delivery it. While the agent is getting the plate, the other agent has nothing to do, which waste some time. Our method learns a more efficiently way, it makes one agent chop both of the tomato and lettuce, meanwhile, the other agent goes to pick up the plate. It makes both of the agents work in parallel, which saves the time.

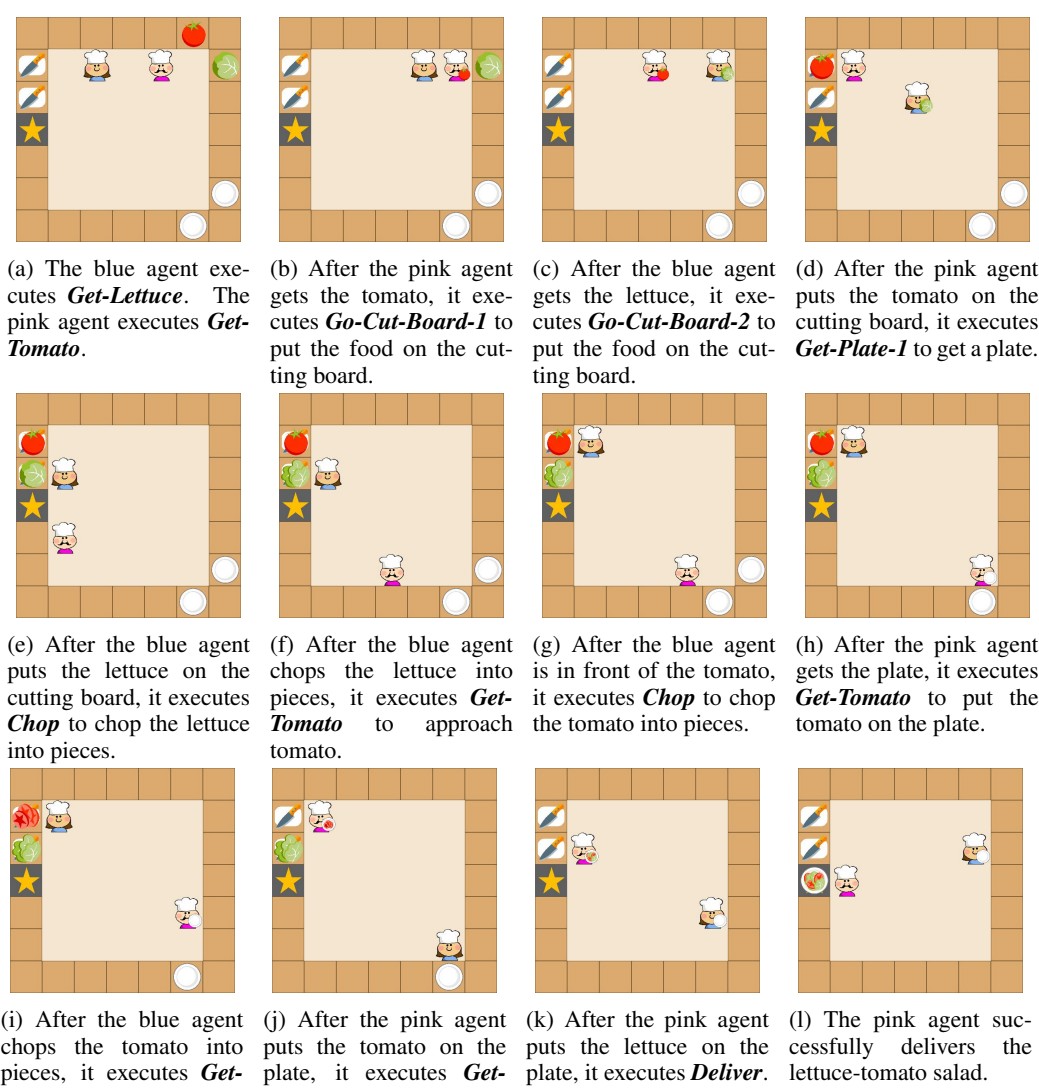

(a) The blue agent executes ***Get-Lettuce***. The pink agent executes ***Get-Tomato***.

(b) After the pink agent gets the tomato, it executes ***Go-Cut-Board-1*** to put the food on the cutting board.

(c) After the blue agent gets the lettuce, it executes ***Go-Cut-Board-2*** to put the food on the cutting board.

(d) After the pink agent puts the tomato on the cutting board, it executes ***Get-Plate-1*** to get a plate.

(e) After the blue agent puts the lettuce on the cutting board, it executes ***Chop*** to chop the lettuce into pieces.

(f) After the blue agent chops the lettuce into pieces, it executes ***Get-Tomato*** to approach tomato.

(g) After the blue agent is in front of the tomato, it executes ***Chop*** to chop the tomato into pieces.

(h) After the pink agent gets the plate, it executes ***Get-Tomato*** to put the tomato on the plate.

(i) After the blue agent chops the tomato into pieces, it executes ***Get-Plate-2*** to avoid blocking pink agent's way.

(j) After the pink agent puts the tomato on the plate, it executes ***Get-Lettuce***.

(k) After the pink agent puts the lettuce on the plate, it executes ***Deliver***.

(l) The pink agent successfully delivers the lettuce-tomato salad.

Figure 20: Visualization of running decentralized policies learned by Mac-IAICC in Overcooked-A.

**Map B.** In this map, the best strategy is that the pink agent should take the advantage of the middle counters to pass vegetables to the other agent. Our method learns a sub-optimal policy such that the blue agent still crosses the narrow passage to get the vegetable at the right side of the map.

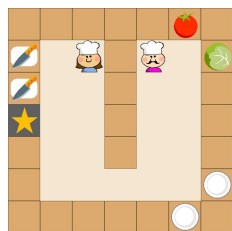
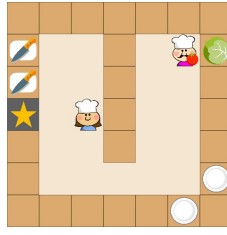
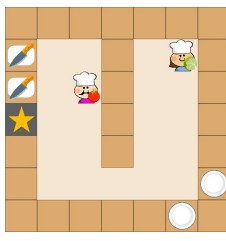
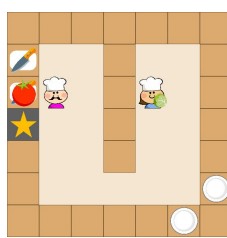

(a) The blue agent executes **Get-Lettuce**. The pink agent executes **Get-Tomato**.

(b) After the pink agent gets the tomato, it executes **Go-Cut-Board-2** to put the food on the cutting board.

(c) After the blue agent gets the lettuce, it executes **Go-Cut-Board-1** to put the food on the cutting board.

(d) After the pink agent puts the tomato on the cutting board, it executes **Chop** to chop the tomato into pieces.

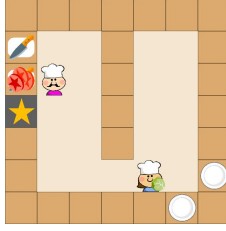
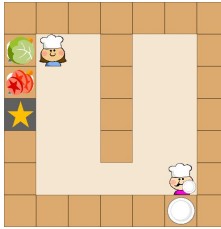
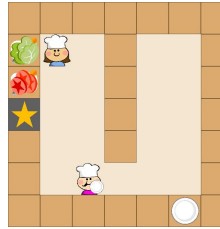
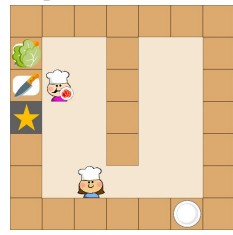

(e) After the pink agent chops the lettuce into pieces, it executes **Get-Plate-1**.

(f) After the blue agent puts the lettuce on the cutting board, it executes **Chop** to chop the lettuce into pieces. After the pink agent gets the plate, it executes **Go-Cut-Board-2** to get the chopped tomato.

(g) After the blue agent chops the lettuce into pieces, it executes **Get-Plate-2** to avoid blocking pink agent's way.

(g) After the pink agent puts the tomato on the plate, it executes **Get-Lettuce**.

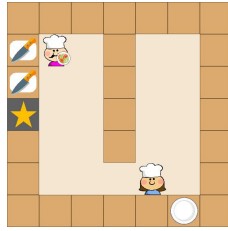
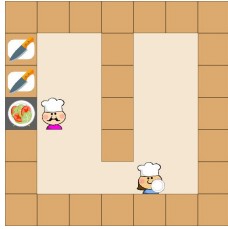

(h) After the pink agent puts the lettuce on the plate, it executes **Deliver**.

(i) The pink agent successfully delivers the lettuce-tomato salad.

Figure 21: Visualization of running decentralized policies learned by Mac-IAICC in Overcooked-B.

**Map C**. In this map, the policy trained by our method learns to use the macro-action ***Go-Counter*** to pass vegetables and plates to teammate.

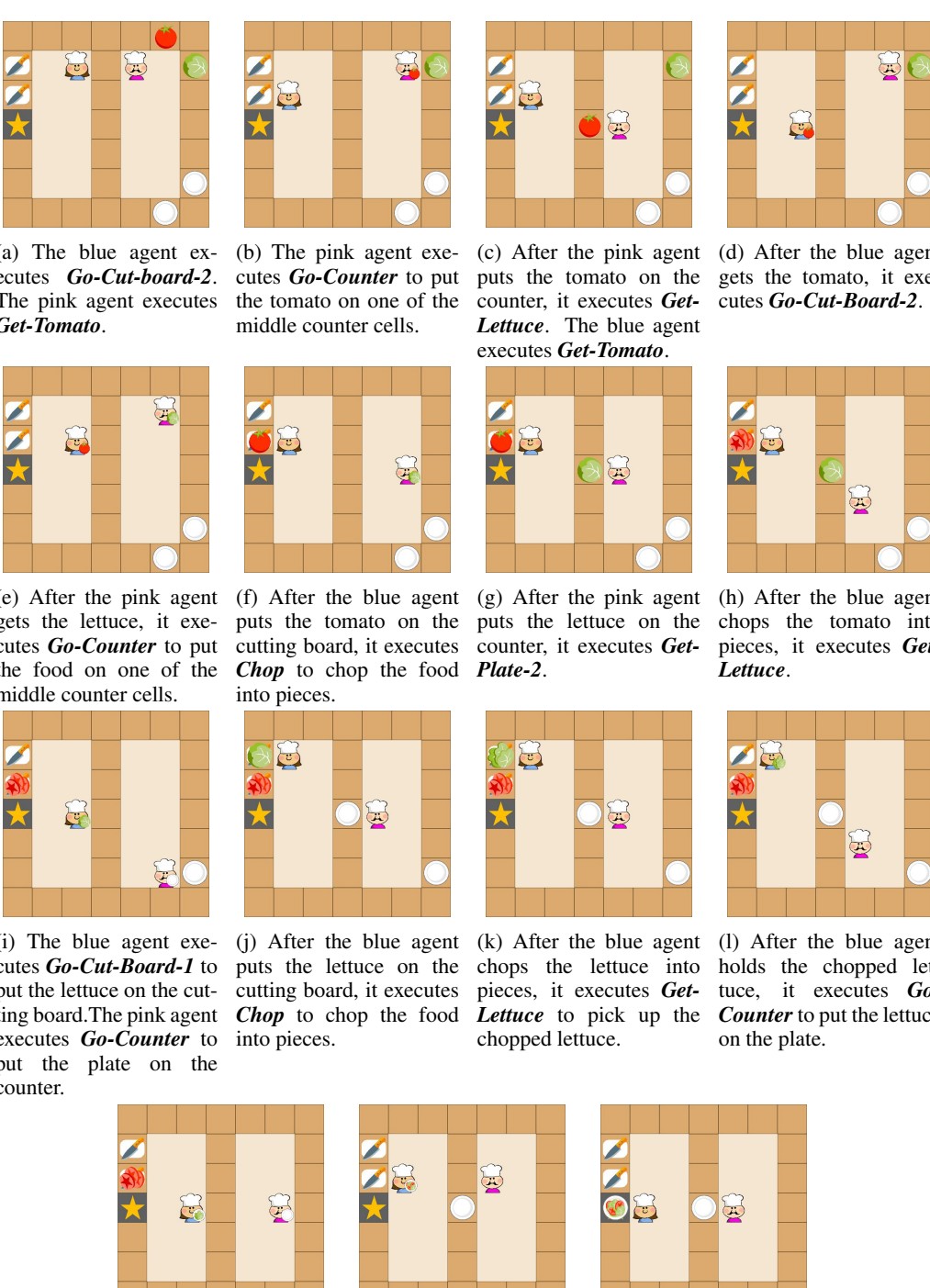

(a) The blue agent executes ***Go-Cut-board-2***. The pink agent executes ***Get-Tomato***.

(b) The pink agent executes ***Go-Counter*** to put the tomato on one of the middle counter cells.

(c) After the pink agent puts the tomato on the counter, it executes ***Get-Lettuce***. The blue agent executes ***Get-Tomato***.

(d) After the blue agent gets the tomato, it executes ***Go-Cut-Board-2***.

(e) After the pink agent gets the lettuce, it executes ***Go-Counter*** to put the food on one of the middle counter cells.

(f) After the blue agent puts the tomato on the cutting board, it executes ***Chop*** to chop the food into pieces.

(g) After the pink agent puts the lettuce on the counter, it executes ***Get-Plate-2***.

(h) After the blue agent chops the tomato into pieces, it executes ***Get-Lettuce***.

(i) The blue agent executes ***Go-Cut-Board-1*** to put the lettuce on the cutting board. The pink agent executes ***Go-Counter*** to put the plate on the counter.

(j) After the blue agent puts the lettuce on the cutting board, it executes ***Chop*** to chop the food into pieces.

(k) After the blue agent chops the lettuce into pieces, it executes ***Get-Lettuce*** to pick up the chopped lettuce.

(l) After the blue agent holds the chopped lettuce, it executes ***Go-Counter*** to put the lettuce on the plate.

(m) After the blue agent puts the lettuce on the plate, it executes ***Get-Tomato***.

(n) After the blue agent puts the tomato on the plate, it executes ***Deliver***.

(o) The blue agent successfully delivers the lettuce-tomato salad.

Figure 22: Visualization of running decentralized policies learned by Mac-IAICC in Overcooked-C.

### A.4.3 WAREHOUSE TOOL DELIVERY

**Warehouse-A**. In this domain, a simple collaborative strategy is that two mobile robots separately assist a particular human (e.g. green robot always delivers tool to W-1 and blue robot always delivers tool to W-2). However, according to each human's working speed, this strategy causes all deliveries are late except the first one such that humans always wait after finishing each subtask, which results in many penalties. By using our method Mac-IAICC, robots learn a more efficient collaboration shown below, which only leads to one delayed delivery but others are all in time.

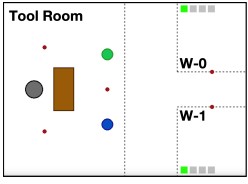

(a) Initial State.

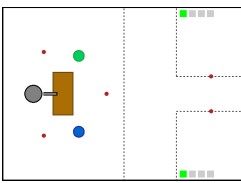

(b) Mobile robots moves towards the table by running **Get-Tool**, and arm robot runs **Search-Tool(0)** to find Tool-0.

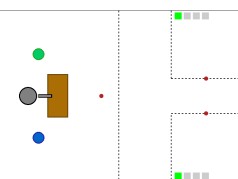

(c) Mobile robots wait there and arm robot keeps looking for the first tool.

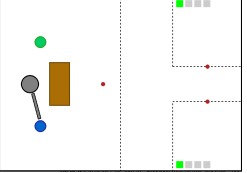

(d) Arm robot executes **Pass-to-M(1)** to pass the first tool to the blue robot.

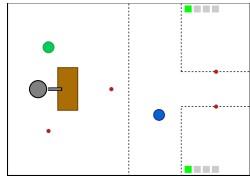

(e) Arm robot executes **Search-Tool(1)** to find Tool-1, and blue robot moves to workshop-1 by executing **Go-W(1)**.

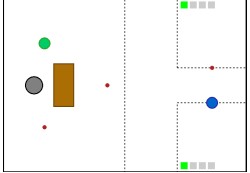

(f) Blue robot successfully delivers Tool-0 to workshop-1.

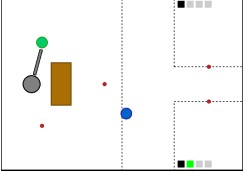

(g) Blue robot runs **Get-Tool** to go back table, and arm robot executes **Pass-to-M(0)** to pass Tool-1 to green robot.

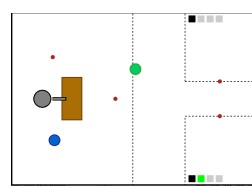

(h) Green robot executes **Go-W(0)** and arm robot runs **Search-Tool(2)**. Blue robot waits for tools.

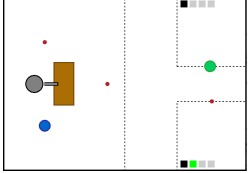

(i) Green observes human-0 needs Tool-0 while it carries Tool-1, so that it executes **Go-W(1)** to check if human-1 needs Tool-1.

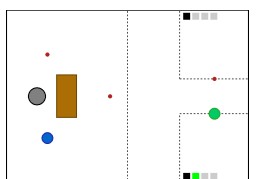

(j) Green robot successfully delivers Tool-1 to human-1.

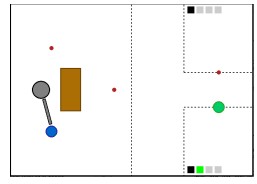

(k) Arm robot executes **Pass-to-M(1)** to pass Tool-2 to blue robot.

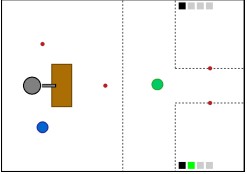

(l) Arm robot executes **Search-Tool(0)** to find the other Tool-0, and green robot runs **Get-Tool** to go back arm robot.

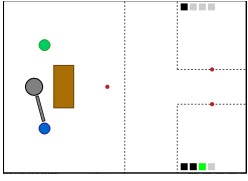

(m) Arm robot executes *Pass-to-M(1)* to pass Tool-0 to blue robot, and blue robot carries Tool-0 and Tool-2 now.

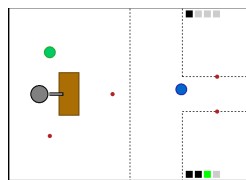

(n) Blue robot smartly runs *Go-W(0)* to first delivery Tool-0 to human-0, and arm robot executes *Search-Tool(1)* to find the other Tool-1.

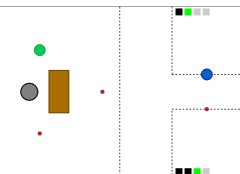

(o) Blue robot successfully delivers Tool-0 to human-0, and human starts working on next subtask.

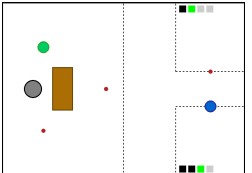

(p) Blue robot executes *Go-W(1)* and successfully delivers Tool-2 to human-1. Human-1 now have obtained all necessary tools.

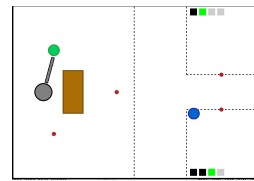

(q) Arm robot executes *Pass-to-M(0)* to pass Tool-1 to green robot, and blue robot runs *Get-Tool* to go back the table.

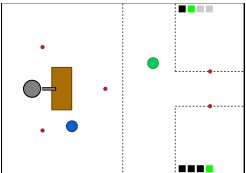

(r) Green robot runs *Go-W(0)* to deliver Tool-1 to human-0, and arm robot executes *Search-Tool(2)* to find the last Tool-2.

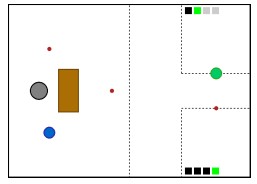

(s) Green robot successfully delivers Tool-1 to human-0.

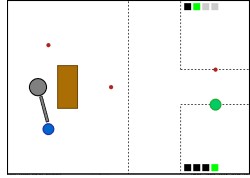

(t) Arm robot executes *Pass-to-M(1)* to give Tool-2 to blue robot, and green robot goes to check human-1 by running *Go-W(1)*.

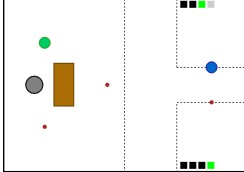

(u) Blue robots directly goes to workshop-0 by running *Go-W(0)* and finishes the last tool delivery for human-0. The entire task is done.

**Warehouse-B**. In this scenario, our hand-coded strategy controls each mobile robot to focus on serving a particular workshop, which is able to finish the task without any delayed delivery. By using Mac-IAICC, robots learn another cooperative behaviors that achieve timely delivery for all tools by using only two mobile robots (the green one and the yellow one) and achieve the same return as the hand-coded one.

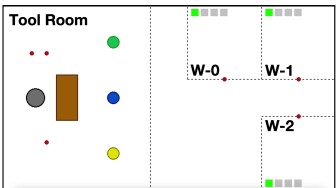

(a) Initial State.

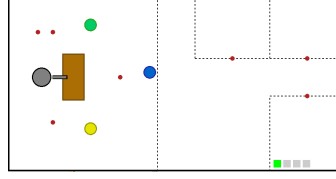

(b) Both green and yellow robots move towards the table by running **Get-Tool**. Blue runs **Go-W(0)** to go to workshop-0. Arm robot runs **Search-Tool(0)** to find Tool-0.

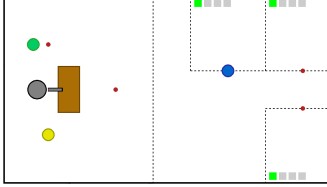

(c) Green and yellow robots wait there and arm robot keeps looking for the Tool-0. Blue robot arrives at workshop-0.

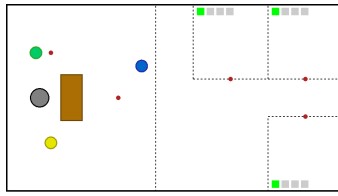

(d) Blue robot runs **Get-Tool** *to go back the table.*

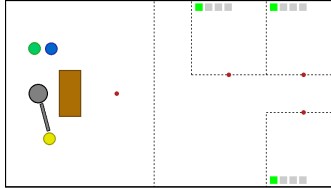

(e) Arm robot executes **Pass-to-M(2)** to pass Tool-0 to yellow robot.

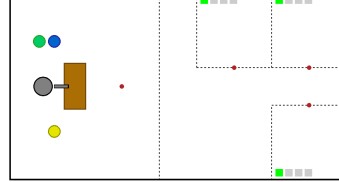

(f) All mobile robots keep running **Get-Tool** to wait there for tools, and arm robot executes **Search-Tool(0)** to find the 2nd Tool-0.

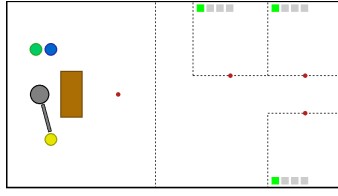

(g) Arm robot executes **Pass-to-M(2)** *to pass the 2nd Tool-0 to the yellow robot, Now yellow robot carries two Tool-O..*

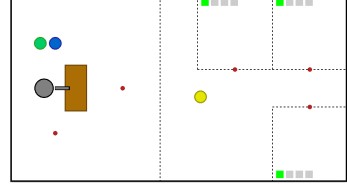

(h) Arm robot executes **Search-Tool(0)** to find the 3rd Tool-0, while yellow robot runs **Go-W(1)** to deliver Tool-0 to the furthest workshop-1.

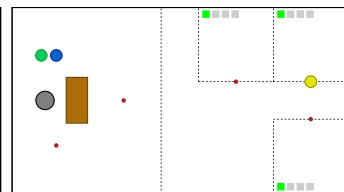

(i) Yellow robot successfully delivers Tool-0 to human-1.

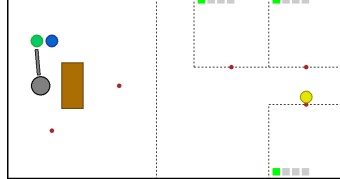

(j) Yellow robot runs **Go-W(2)** to deliver the other Tool-0 to human-2, and arm robot executes **Pass-to-M(0)** to pass the 3rd Tool-O to green robot.

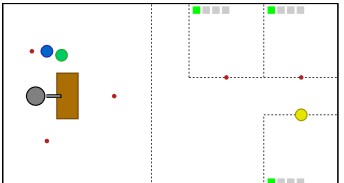

(k) Yellow robot successfully delivers Tool-0 to human-2. Green robot runs **Go-W(0)** to deliver Tool-0 to human-0. Arm robot executes **Search-Tool(1)** to find Tool-1.

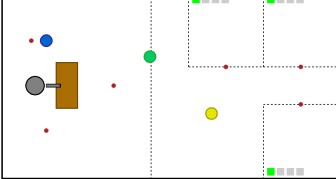

(l) Yellow robot runs **Get-Tool** to return tool room. Green robot keeps moving towards workshop-0 under the execution of **Go-W(0)**. Arm robot is looking for Tool-0 under the running of previous macro-action.

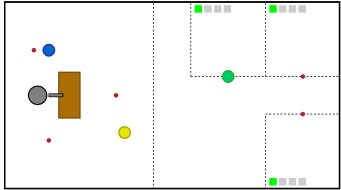

(m) Arm robot executes **Search-Tool(1)** again to find the 2nd Tool-1. Green robot successfully delivers Tool-0 to human-0. Yellow robot is moving towards table.

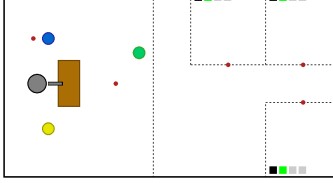

(n) Green robot runs **Get-Tool** to go back table. Yellow robot is waiting for tools.

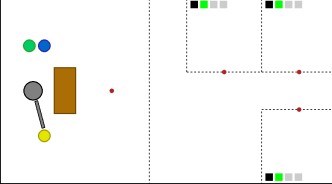

(o) Arm robot executes **Pass-to-M(2)** to pass a Tool-1 to yellow robot.

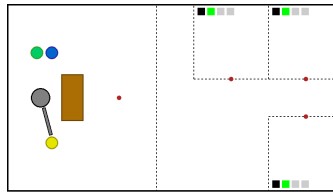

(p) Arm robot executes **Pass-to-M(2)** again to pass the other Tool-1 to yellow robot.

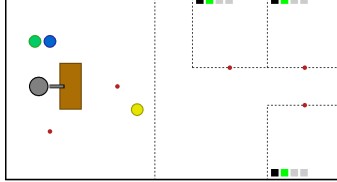

(q) Arm robot executes **Search-Tool(1)** to find the 3rd Tool-1, and yellow robot runs **Go-W(1)** to first deliver Tool-1 to the furthest workshop-1.

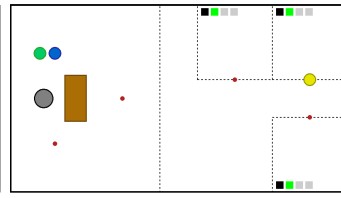

(r) Yellow robot successfully delivers Tool-1 to human-1.

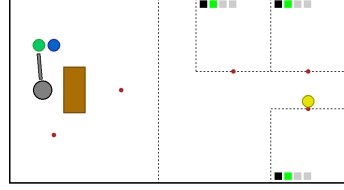

(s) Yellow robot runs **Go-W(2)** to deliver the other Tool-1 to human-2, and arm robot executes **Pass-to-M(0)** to pass the 3rd Tool-1 to green robot.

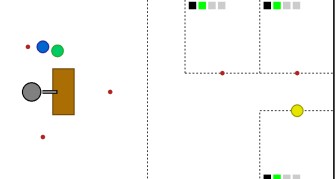

(t) Yellow robot successfully delivers Tool-1 to human-2. Green robot runs **Go-W(0)** to deliver Tool-1 to human-0. Arm robot executes **Search-Tool(2)** to find Tool-2.

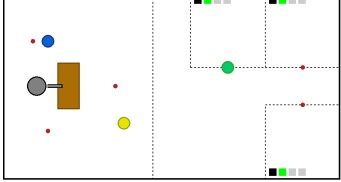

(u) Arm robot executes **Search-Tool(2)** again to find the 2nd Tool-2. Green robot successfully delivers Tool-1 to human-0. Yellow robot is moving towards table.

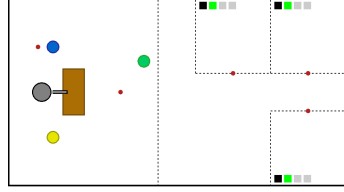

(v) Green robot runs **Get-Tool** to go back table. Yellow robot is waiting for tools.

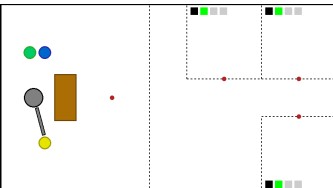

(w) Arm robot executes **Pass-to-M(2)** to pass a Tool-2 to yellow robot.

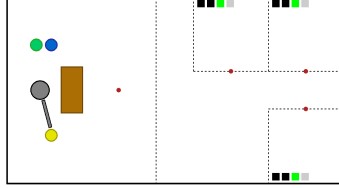

(x) Arm robot executes **Pass-to-M(2)** again to pass the other Tool-2 to yellow robot.

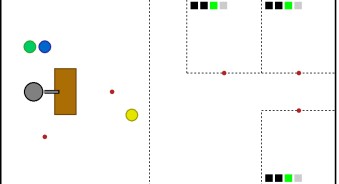 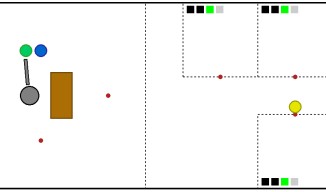

(y) Arm robot executes *Search-Tool(2)* to find the 3rd Tool-2, and yellow robot runs *Go-W(1)* to first deliver Tool-2 to the furthest workshop-1.

(z) Yellow robot successfully delivers Tool-2 to human-1.

(A) Yellow robot runs *Go-W(2)* to deliver the other Tool-2 to human-2, and arm robot executes *Pass-to-M(0)* to pass the 3rd Tool-2 to green robot.

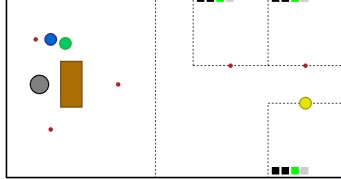 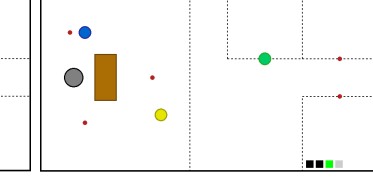

(B) Yellow robot successfully delivers Tool-2 to human-2. Green robot runs *Go-W(0)* to deliver Tool-2 to human-0.

(C) Green robot successfully delivers Tool-2 to human-2. Humans have received all tools, and for robots, the task is done.

**Warehouse-C**

In this scenario, the arm robot leans an efficient order to find correct tools that humans need in such a way that first getting the proper tools for each human's second subtask; second, getting the proper tools for each human's third subtask; and finally getting the proper tools for each human's last subtask. Mobile robots are also clever such that the green robot focuses on delivering tools to two workshops (0 and 2) and gives the priority to the faster human in workshop-0, meanwhile, the blue robot mainly focuses on assisting the human-1 in workshop-1 (W-1) by delivering the correct tools one by one in time, but also helps delivering one tool to human-2.

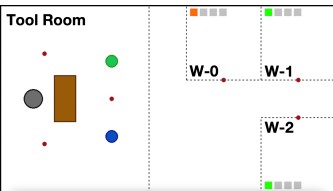 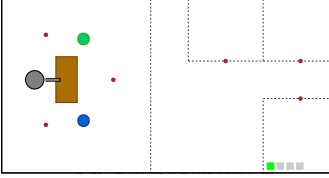 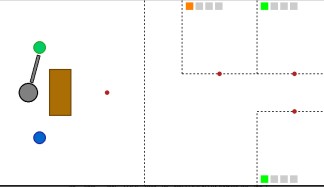

(a) Initial State.

(b) Arm robot executes **Search-Tool(0)** to find Tool-0, and two mobile robots are moving towards the table by running **Get-Tool**.

(c) Arm robot executes **Pass-to-M(0)** to pass Tool-0 to green robot. Blue robot waits there for tools.

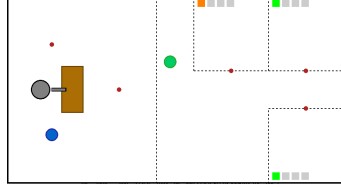 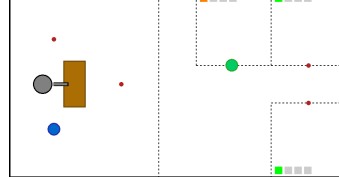 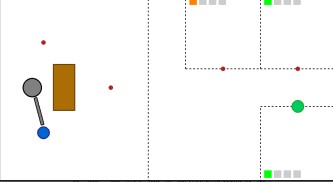

(d) Green robot runs **Go-W(0)** to deliver Tool-0 to the faster human, and arm robot executes **Search-Tool(0)** to find the 2nd Tool-0.

(e) Green robot successfully delivers Tool-0 to human-0, and arm robot is still under the execution of previous macro-action.

(f) Arm robot executes **Pass-to-M(1)** to give Tool-0 to blue robot, and green robot goes to check human-2's status by running **Go-W(2)**.

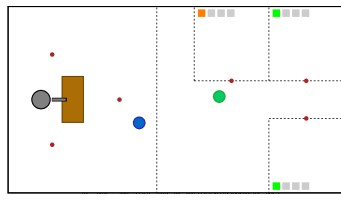 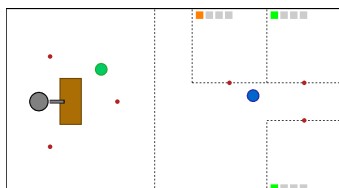 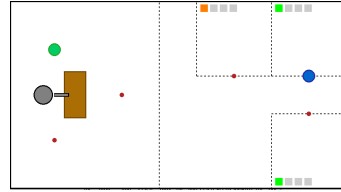

(g) Arm robot executes **Search-Tool(0)** to find the 3rd Tool-0. Blue robot moves towards workshop-1 by running **Go-W(1)** and green robot runs **Get-Tool** to go back arm robot.

(h) Arm robot executes **Search-Tool(1)** to find Tool-1. Mobile robots continue running the previous macro-actions.

(i) Blue robot successfully delivers Tool-0 to human-1, and green robot waits there for next tool. Arm robot is under the execution of previous macro-action.

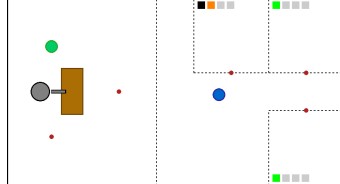 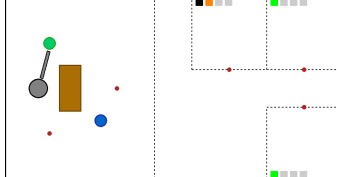 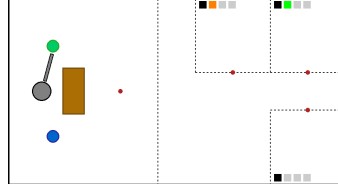

(j) Blue robot goes back table by running **Get-Tool**.

(k) Arm robot passes the last Tool-0 to the green robot by executing **Pass-to-M(0)**, and blue robot is under the execution of **Get-Tool**.

(l) Arm robot executes **Pass-to-M(0)** again to give Tool-1 to green robot, and blue arrives at table and waits for next tool.

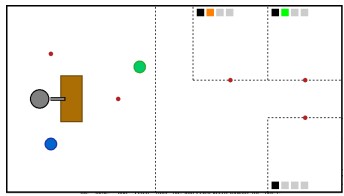

(m) Green robot runs *Go-W(0)* to first deliver Tool-1 to the faster human, and arm robot executes *Search-Tool(1)* to find the 2nd Tool-1.

(n) Green robot successfully delivers Tool-1 to human-0, and arm robot is still searching for Tool-1.

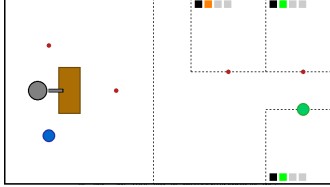

(o) Arm robot executes *Search-Tool(1)* again to find the last Tool-1, meanwhile, green robot successfully delivers Tool-0 to human-2 by running *Go-W(2)*.

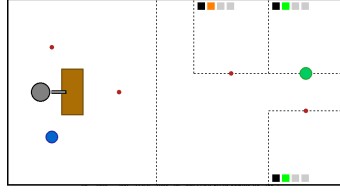

(p) Green robot runs *Go-W(1)* to check the status of human-1.

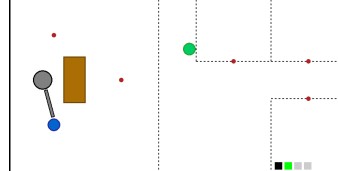

(q) Arm robot executes *Pass-to-M(1)* to give Tool-1 to blue robot, and green robot runs *Get-Tool* to go back table.

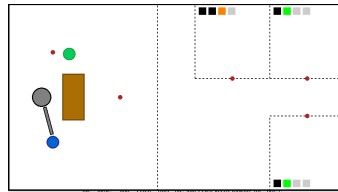

(r) Arm robot executes *Pass-to-M(1)* again to give the last Tool-1 to blue robot, and now blue robot carries two Tool-1s.

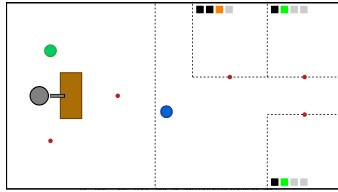

(s) Arm robot executes *Search-Tool(2)* to find Tool-2. Green robot waits there for next tool, and blue robot moves towards workshop-1 by running *Go-W(1)* for delivery.

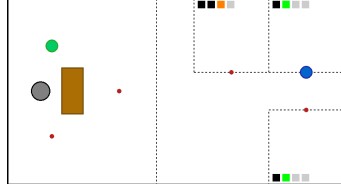

(t) Blue robot delivers Tool-1 to human-1 in time.

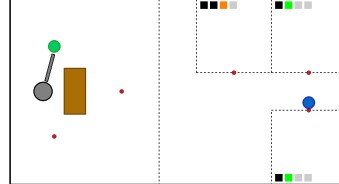

(u) Arm robot executes *Pass-to-M(0)* to give Tool-2 to green robot, and blue robot runs *Go-W(2)* to send Tool-1 to workshop-2.

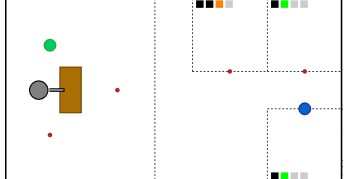

(v) Arm robot executes *Search-Tool(2)* to find the 2nd Tool-2. Green robot keeps waiting there for more tools by running *Get-Tool* again, and blue robot delivers Tool-1 to human-2 in time.

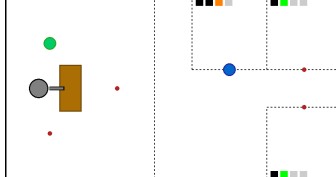

(w) Blue robot arrives at workshop-0 to check the status of human-0 by running *Go-W(0)*, and arm robot is still looking for the 2nd Tool-2.

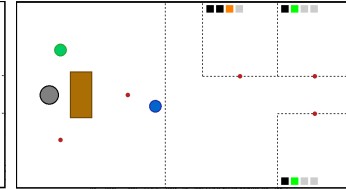

(x) Blue robot goes back tool room by running *Get-Tool*, and arm robot is under the execution of *Pass-to-M(0)*.

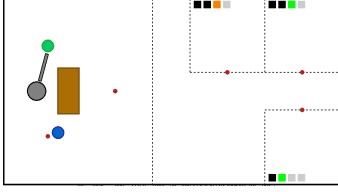 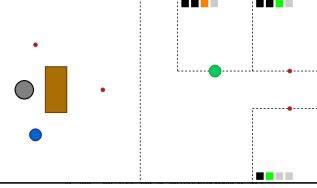

(y) Arm robot passes the 2nd Tool-2 to green robot by executing **Pass-to-M(0)**, and now green robot carries two Tool-2s. Blue robot is under the execution of previous macro-action.

(z) Green robot runs **Go-W(0)** to first send Tool-2 to the faster human-0, and blue robot waits beside table for next tool. Arm robot executes **Search-Tool(2)** to find the last Tool-2.

(A) Green robot delivers Tool-2 to human-0 in time.

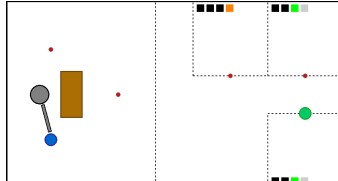 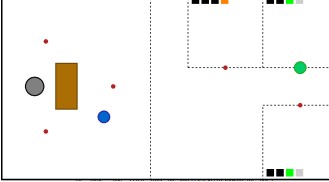 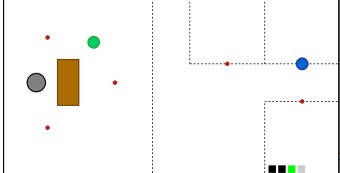

(B) Arm robot passes the last Tool-2 to blue robot by executing **Pass-to-M(1)**. Green arrives at workshop-2 right after human-2 finishes the second subtask and is going to start working on the next subtask. According to the dynamics setup, at this timestep, human-2 cannot possess the Tool-2 from the green robot.

(C) Green robot then goes to workshop-1 to send Tool-2 to human-1 by executing **Go-W(1)** and successfully delivers it to human-1. Blue robot cannot observes this so that it still executes **Go-W(1)** to send Tool-2 to human-1.

(D) Blue robot arrives at workshop-1 and observes that human-1 has already obtained the last tool he needs. Blue robot decides to send the Tool-2 to human-2 by running **Go-W(2)**.

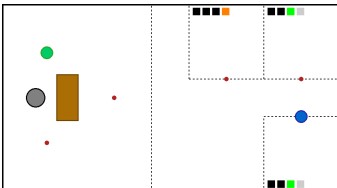

(E) Blue robot successfully delivers Tool-2 to human-2. Humans have received all tools, and for robots, the task is done.

## A.5 HYPER-PARAMETERS

In following subsections, we first list the hyper-parameter candidates used for tuning each method via grid search in the corresponding domain, and then show the hyper-parameter table with the parameters used by each method achieving the best performance. We choose the best performance of each method depending on its final converged value as the first priority and the sample efficiency as the second.

### A.5.1 BOX PUSHING

Table 1: Hyper-parameter candidates for grid search tuning.

| | |
|---|---|
| learning rate pair (actor,critic) | (1e-3,3e-3), (1e-3,1e-3) (5e-4,3e-3), (5e-4,1e-3) (5e-4,5e-4), (3e-4,3e-3) |
| Episodes per train | 8, 16, 32 |
| Target-net update freq (episode) | 32, 64, 128 |
| N-step TD | 0, 3, 5 |

Table 2: Hyper-parameter candidates for grid search tuning.

| | |
|---|---|
| learning rate pair (actor,critic) | (1e-3,3e-3), (1e-3,1e-3) (5e-4,3e-3), (5e-4,1e-3) (5e-4,5e-4), (3e-4,3e-3) |
| Episodes per train | 48 |
| Target-net update freq (episode) | 48, 96, 144 |
| N-step TD | 0, 3, 5 |

Table 3: Hyper-parameters used for methods in Box Pushing $6 \times 6$.

| Parameter | IAC | CAC | Mac-IAC | Mac-CAC | Mac-NIACC | Mac-IAICC |
|---|---|---|---|---|---|---|
| Training Episodes | 40K | 40K | 40K | 40K | 40K | 40K |
| Actor learning rate | 0.001 | 0.0005 | 0.0005 | 0.001 | 0.0005 | 0.0003 |
| Critic learning rate | 0.003 | 0.001 | 0.003 | 0.003 | 0.001 | 0.003 |
| Episodes per train | 8 | 8 | 32 | 32 | 32 | 32 |
| Target-net update freq (episode) | 32 | 128 | 32 | 64 | 32 | 128 |
| N-step TD | 5 | 5 | 0 | 3 | 0 | 0 |
| $\epsilon_{start}$ | 1 | 1 | 1 | 1 | 1 | 1 |
| $\epsilon_{end}$ | 0.01 | 0.01 | 0.01 | 0.01 | 0.01 | 0.01 |
| $\epsilon_{decay}$ (episode) | 4K | 4K | 4K | 4K | 4K | 4K |

Table 4: Hyper-parameters used for methods in Box Pushing $8 \times 8$.

| Parameter | IAC | CAC | Mac-IAC | Mac-CAC | Mac-NIACC | Mac-IAICC |
|---|---|---|---|---|---|---|
| Training Episodes | 40K | 40K | 40K | 40K | 40K | 40K |
| Actor learning rate | 0.001 | 0.001 | 0.0005 | 0.0003 | 0.0005 | 0.0003 |
| Critic learning rate | 0.003 | 0.003 | 0.003 | 0.003 | 0.001 | 0.003 |
| Episodes per train | 8 | 8 | 48 | 32 | 32 | 32 |
| Target-net update freq (episode) | 32 | 128 | 48 | 32 | 128 | 64 |
| N-step TD | 3 | 3 | 0 | 3 | 0 | 0 |
| $\epsilon_{start}$ | 1 | 1 | 1 | 1 | 1 | 1 |
| $\epsilon_{end}$ | 0.01 | 0.01 | 0.01 | 0.01 | 0.01 | 0.01 |
| $\epsilon_{decay}$ (episode) | 4K | 4K | 4K | 4K | 4K | 4K |

Table 5: Hyper-parameters used for methods in Box Pushing $10 \times 10$.

| Parameter | IAC | CAC | Mac-IAC | Mac-CAC | Mac-NIACC | Mac-IAICC |
|---|---|---|---|---|---|---|
| Training Episodes | 40K | 40K | 40K | 40K | 40K | 40K |
| Actor learning rate | 0.001 | 0.001 | 0.001 | 0.0005 | 0.0005 | 0.0003 |
| Critic learning rate | 0.003 | 0.003 | 0.001 | 0.001 | 0.001 | 0.003 |
| Episodes per train | 8 | 8 | 16 | 32 | 32 | 32 |
| Target-net update freq (episode) | 64 | 128 | 32 | 128 | 128 | 32 |
| N-step TD | 0 | 0 | 3 | 3 | 0 | 0 |
| $\epsilon_{start}$ | 1 | 1 | 1 | 1 | 1 | 1 |
| $\epsilon_{end}$ | 0.01 | 0.01 | 0.01 | 0.01 | 0.01 | 0.01 |
| $\epsilon_{decay}$ (episode) | 6K | 6K | 6K | 6K | 6K | 6K |

Table 6: Hyper-parameters used for methods in Box Pushing $12 \times 12$.

| Parameter | IAC | CAC | Mac-IAC | Mac-CAC | Mac-NIACC | Mac-IAICC |
|---|---|---|---|---|---|---|
| Training Episodes | 40K | 40K | 40K | 40K | 40K | 40K |
| Actor learning rate | 0.001 | 0.001 | 0.001 | 0.001 | 0.0005 | 0.0003 |
| Critic learning rate | 0.003 | 0.003 | 0.003 | 0.003 | 0.0005 | 0.003 |
| Episodes per train | 8 | 8 | 16 | 8 | 48 | 48 |
| Target-net update freq (episode) | 128 | 128 | 128 | 128 | 48 | 144 |
| N-step TD | 0 | 0 | 3 | 0 | 0 | 0 |
| $\epsilon_{start}$ | 1 | 1 | 1 | 1 | 1 | 1 |
| $\epsilon_{end}$ | 0.01 | 0.01 | 0.01 | 0.01 | 0.01 | 0.01 |
| $\epsilon_{decay}$ (episode) | 6K | 6K | 6K | 6K | 6K | 6K |

Table 7: Hyper-parameters used for methods in Box Pushing $14 \times 14$.

| Parameter | IAC | CAC | Mac-IAC | Mac-CAC | Mac-NIACC | Mac-IAICC |
|---|---|---|---|---|---|---|
| Training Episodes | 40K | 40K | 40K | 40K | 40K | 40K |
| Actor learning rate | 0.001 | 0.001 | 0.001 | 0.001 | 0.0005 | 0.0005 |
| Critic learning rate | 0.003 | 0.003 | 0.001 | 0.001 | 0.0005 | 0.003 |
| Episodes per train | 8 | 8 | 8 | 16 | 32 | 48 |
| Target-net update freq (episode) | 128 | 128 | 32 | 128 | 128 | 48 |
| N-step TD | 0 | 0 | 3 | 3 | 0 | 0 |
| $\epsilon_{start}$ | 1 | 1 | 1 | 1 | 1 | 1 |
| $\epsilon_{end}$ | 0.01 | 0.01 | 0.01 | 0.01 | 0.01 | 0.01 |
| $\epsilon_{decay}$ (episode) | 8K | 8K | 8K | 8K | 8K | 8K |

A.5.2 OVERCOOKED

Table 8: Hyper-parameter candidates for grid search tuning.

| | |
|---|---|
| learning rate pair (actor,critic) | (1e-4, 3e-3) (3e-4,3e-3) |
| Episodes per train | 4 |
| Target-net update freq (episode) | 8, 16, 32 |
| N-step TD | 3, 5 |

Table 9: Hyper-parameter candidates for grid search tuning.

| | |
|---|---|
| learning rate pair (actor,critic) | (1e-4, 3e-3) (3e-4,3e-3) |
| Episodes per train | 8, 16 |
| Target-net update freq (episode) | 16, 32, 64 |
| N-step TD | 3, 5 |

Table 10: Hyper-parameters used for methods in Overcooked-A.

| Parameter | IAC | CAC | Mac-IAC | Mac-CAC | Mac-NIACC | Mac-IAICC |
|---|---|---|---|---|---|---|
| Training Episodes | 100K | 100K | 100K | 100K | 100K | 100K |
| Actor learning rate | 0.0003 | 0.0003 | 0.0003 | 0.0001 | 0.0003 | 0.0003 |
| Critic learning rate | 0.003 | 0.003 | 0.003 | 0.003 | 0.003 | 0.003 |
| Episodes per train | 16 | 4 | 16 | 4 | 8 | 8 |
| Target-net update freq (episode) | 64 | 16 | 32 | 8 | 16 | 32 |
| N-step TD | 5 | 5 | 5 | 5 | 5 | 5 |
| $\epsilon_{start}$ | 1 | 1 | 1 | 1 | 1 | 1 |
| $\epsilon_{end}$ | 0.05 | 0.05 | 0.05 | 0.05 | 0.05 | 0.05 |
| $\epsilon_{decay}$ (episode) | 40K | 40K | 40K | 40K | 40K | 40K |

Table 11: Hyper-parameters used for methods in Overcooked-B.

| Parameter | IAC | CAC | Mac-IAC | Mac-CAC | Mac-NIACC | Mac-IAICC |
|---|---|---|---|---|---|---|
| Training Episodes | 100K | 100K | 100K | 100K | 100K | 100K |
| Actor learning rate | 0.0003 | 0.0001 | 0.0003 | 0.0001 | 0.0003 | 0.0003 |
| Critic learning rate | 0.003 | 0.003 | 0.003 | 0.003 | 0.003 | 0.003 |
| Episodes per train | 8 | 8 | 8 | 8 | 4 | 4 |
| Target-net update freq (episode) | 32 | 16 | 64 | 16 | 32 | 8 |
| N-step TD | 5 | 5 | 5 | 3 | 5 | 3 |
| $\epsilon_{start}$ | 1 | 1 | 1 | 1 | 1 | 1 |
| $\epsilon_{end}$ | 0.05 | 0.05 | 0.05 | 0.05 | 0.05 | 0.05 |
| $\epsilon_{decay}$ (episode) | 40K | 40K | 40K | 40K | 40K | 40K |

Table 12: Hyper-parameters used for methods in Overcooked-C.

| Parameter | IAC | CAC | Mac-IAC | Mac-CAC | Mac-NIACC | Mac-IAICC |
|---|---|---|---|---|---|---|
| Training Episodes | 100K | 100K | 100K | 100K | 100K | 100K |
| Actor learning rate | 0.0003 | 0.0003 | 0.0003 | 0.0001 | 0.0003 | 0.0003 |
| Critic learning rate | 0.003 | 0.003 | 0.003 | 0.003 | 0.003 | 0.003 |
| Episodes per train | 16 | 8 | 16 | 4 | 16 | 4 |
| Target-net update freq (episode) | 64 | 32 | 32 | 32 | 64 | 32 |
| N-step TD | 5 | 5 | 5 | 3 | 5 | 5 |
| $\epsilon_{start}$ | 1 | 1 | 1 | 1 | 1 | 1 |
| $\epsilon_{end}$ | 0.05 | 0.05 | 0.05 | 0.05 | 0.05 | 0.05 |
| $\epsilon_{decay}$ (episode) | 40K | 40K | 40K | 40K | 40K | 40K |

### A.5.3 WAREHOUSE TOOL DELIVERY

Table 13: Hyper-parameter candidates for grid search tuning.

| | |
|---|---|
| learning rate pair (actor,critic) | (1e-3,1e-3), (5e-4,1e-3) (5e-4,5e-4) (3e-4,3e-3) |
| Episodes per train | 4, 8 |
| Target-net update freq (episode) | 8, 16, 32, 64 |
| N-step TD | 0, 3, 5 |

Table 14: Hyper-parameter candidates for grid search tuning.

| | |
|---|---|
| learning rate pair (actor,critic) | (1e-3,1e-3), (5e-4,1e-3) (5e-4,5e-4) (3e-4,3e-3) |
| Episodes per train | 16 |
| Target-net update freq (episode) | 16, 32, 64 |
| N-step TD | 0, 3, 5 |

Table 15: Hyper-parameters used for methods in Warehouse-A.

| Parameter | Mac-IAC | Mac-CAC | Mac-NIACC | Mac-IAICC |
|---|---|---|---|---|
| Training Episodes | 40K | 40K | 40K | 40K |
| Actor learning rate | 0.0005 | 0.0003 | 0.0003 | 0.0005 |
| Critic learning rate | 0.0005 | 0.003 | 0.003 | 0.0005 |
| Episodes per train | 16 | 4 | 8 | 4 |
| Target-net update freq (episode) | 64 | 16 | 32 | 64 |
| N-step TD | 5 | 3 | 5 | 5 |
| $\epsilon_{start}$ | 1 | 1 | 1 | 1 |
| $\epsilon_{end}$ | 0.01 | 0.01 | 0.01 | 0.01 |
| $\epsilon_{decay}$ (episode) | 10K | 10K | 10K | 10K |

Table 16: Hyper-parameters used for methods in Warehouse-B.

| Parameter | Mac-IAC | Mac-CAC | Mac-NIACC | Mac-IAICC |
|---|---|---|---|---|
| Training Episodes | 90K | 90K | 90K | 90K |
| Actor learning rate | 0.0005 | 0.0003 | 0.0003 | 0.0003 |
| Critic learning rate | 0.0005 | 0.003 | 0.003 | 0.003 |
| Episodes per train | 8 | 8 | 4 | 4 |
| Target-net update freq (episode) | 32 | 64 | 32 | 16 |
| N-step TD | 5 | 3 | 5 | 5 |
| $\epsilon_{start}$ | 1 | 1 | 1 | 1 |
| $\epsilon_{end}$ | 0.01 | 0.01 | 0.01 | 0.01 |
| $\epsilon_{decay}$ (episode) | 10K | 10K | 10K | 10K |

Table 17: Hyper-parameters used for methods in Warehouse-C.

| Parameter | Mac-IAC | Mac-CAC | Mac-NIACC | Mac-IAICC |
|---|---|---|---|---|
| Training Episodes | 90K | 90K | 90K | 90K |
| Actor learning rate | 0.0005 | 0.0005 | 0.0003 | 0.0003 |
| Critic learning rate | 0.001 | 0.001 | 0.003 | 0.003 |
| Episodes per train | 4 | 4 | 4 | 4 |
| Target-net update freq (episode) | 64 | 64 | 32 | 64 |
| N-step TD | 5 | 3 | 5 | 5 |
| $\epsilon_{start}$ | 1 | 1 | 1 | 1 |
| $\epsilon_{end}$ | 0.05 | 0.05 | 0.05 | 0.05 |
| $\epsilon_{decay}$ (episode) | 10K | 10K | 10K | 10K |

