# OpenReview forum: "Asynchronous Multi-Agent Actor-Critic with Macro-Actions"
_ICLR.cc/2022/Conference — ICLR 2022 Submitted_

### Official Review · Reviewer_Exb6 · 2021-10-31

**Correctness:** 3
**Technical Novelty And Significance:** 3
**Empirical Novelty And Significance:** 3
**Recommendation:** 5
**Confidence:** 3

**Main Review:**

The paper gives a relatively clear definition of the domain of interest and its value/applicability. The algorithms are mostly clear and seem sensible. There is some significant novelty in the use of central critics with independent gradients particularly when those central critics have to be individualised. The need for centralisation does limit this in the number of agents that can be coordinated though and this could be more clearly commented on. The experiments comparing macro-action based methods are clear and well interpreted with some interesting findings. Finally, for the strengths, I do think that these methods may have some viability for use in industrial applications.

There are a couple of things that I think weaken the paper. The first of these is quite general and relates to the framework in which all this is being done. There seems to be an unusual asymetry to how the two levels of the action hierarchy are treated. There are different policies of course, but the action sets are strictly macro actions, which means that all primitive action decisions must be part of an option. This has implications for generalisation across the same task or from one task to another, as it forces the option to contain both reusable skill information as well as global task information. More surprising to me, observations are split into low level and high level and I cannot see the justification for that. One consequence of this asymmetry emerges in the tested environments. For instance the box pushing environment requires macro actions of length 1 to be defined, which mimic three of the primitive actions.

A second weakness of the paper relates to the ambiguity in some of the descriptions. In particular, the definition of squeezed trajectories and how these are incorporated into each of the proposed algorithms could be much clearer. Another thing that could be clearer is that this paper presents methods for the macro-actions and how to induce coordination with these. It doesn't address how the options might be learned or developed, nor how such things as initiation sets and termination conditions should be defined - a non-trivial thing see (Jong et al., 2008) . As such there is a serious gap between what is proposed here and the use of these methods in practice. See (Konidaris and Barto, 2007; Bacon et al. , 2017) for examples of works which might address this concern.

Finally, there is a weakness in the experiments in terms of fair comparison. In the experiments, there is quite a lot of pre-encoded knowledge in the macro actions. Not least the one-step macro actions, but also examples like: the get-tomato-like actions in the Overcooked domain, or the Move-to-big-box action in the Box Pushing. It seems to me that these options are already predefined as deterministic policies, as no reward structure is given to these nor is there a mechanism for propagating global rewards back in time to relevant option executions (a better approach). Am I right that these macro action methods are given their options at the start? In which case, the comparison between macro-action based methods (Mac-CAC, Mac-IACC and Mac-IAICC) and primitive action based methods (IAC and IACC) would be demonstrably biased towards the macro-action based methods. I would argue that this knowledge injection is at least partly responsible for the big difference between the two types of methods seen in Figure 3. A fairer comparison would be to have options learned as they go for the macro-action methods.

## Notes

Some more detailed notes/comments are as follows:

In Sec 2, what is mathbb H in the definitions? And why does this relate to finite horizon? Ah, is it an integer? This could be clearer.

The authors state that:
> asynchronicity of macro-action execution over agents, where agents may start and complete their own macro-actions at different time-steps

But argue that:
> Previous deep MARL methods for Dec-POMDPs cannot work in this case as they are all based on primitive actions synchronously executed by agents.

However the formulism given in section 2 does model joint actions at each time step which are cross products over all agents in the task. So the primitive actions are taken synchronously (at least as treated by the environment). This should be clarified.

The explanation of squeezed trajectories in Sections 2.5 and 3 are not clear enough. I would recommend the authors include some mathematical equations or definitions to make this clearer.

In Equation (8) it's not clear how the $\vec{h}$ and $\vec{m}$ objects are constructed. As argued already, the macro actions are initiated and terminated asynchronously but the $\vec{m}$ is a joint macro action, so what is that? For instance, at primitive time $t$, does the joint macro action use all the options that agents are in, even if they initiated this some steps ago? How does this asynchronicity influence the convergence properties of the critic and policy gradient optimisation?

It seems odd that you need to explicitly manufacture exploration decay. You say:
> Exploration is deployed by applying a linear decaying $\epsilon$-soft policy.

But the actor-critic paradigm is one which, it is argued, controls the exploration/exploitation trade-off. This should really be justified more clearly in the paper.

## References
(Jong et al., 2008) Nicholas K. Jong Todd Hester Peter Stone (2008). The Utility of Temporal Abstraction in Reinforcement Learning. AAMAS 08.
(Konidaris and Barto, 2007) George Konidaris and Andrew Barto (2007). Building Portable Options: Skill Transfer in Reinforcement Learning.
(Sutton et al., 1999) Richard Sutton, Doina Precup, and Satinder Singh (1999). Between MDPs and Semi-MDPs: A Framework for Temporal Abstraction in Reinforcement Learning. Artificial Intelligence 112:181–211.
(Bacon et al. , 2017)  Pierre-Luc Bacon, Jean Harb and Doina Precup (2017). The Option-Critic Architecture. AAAI 2017.

**Summary Of The Paper:**

The paper presents a method for learning policy gradient based methods on macro-actions in environments where multiple agents initiate and terminate actions at different timesteps. The paper identifies a suitable framework (MacDec-POMDPs) for these tasks and develops a series of actor-critic mechanisms to allow agents to coordinate asynchronous macros actions. Four main macro-action algorithms are proposed:
* Mac-IAC - that applies a variance reduced version of actor critic directly to the multi-agent domain
* Mac-CAC - which treats all agents as a single global agent
* Mac-IACC - described as naive, which uses a centralised critic but derives independent policy gradients for each agent
* Mac-IAICC - which again uses a centralised critic like mechanism but one where the termination times of the individual agent correct the traces used to train this central critic. Again this derives independent policy gradients for each agent.

The experimental section presents three models: box pushing, overcooked and warehouse. From my reading of the paper, these models provide predefined options to all agents (so all agents have access to all options), where some options are extended and some are single step. Rewards are defined that are broadcast to all agents, and are used to learn the coordination policy (high level) but not the low level options, in the case of the macro action algorithms. Two non-macro action methods are evaluated against Mac-IAC and Mac-CAC on Box-pushing and Overcooked. This finds that macro-action methods are better than the primitive action methods (less interesting - see notes) and that there are situations in which Mac-IAC is better but more typically Mac-CAC outperforms Mac-IAC when there is a significant need for coordination. The next series of experiments compares the four macro-actions methods and show (most pertinantly) that the Mac-IAICC method achieves better coordination than Mac-IAC, nearing the performance of Mac-CAC even in the most coordination intensive conditions. Naive Mac-IACC improves upon Mac-IAC in terms of coordination but suffers in coordination intensive tasks. This supports the authors' argument that the variance in the shared critic for Naive Mac-IACC is damaging and that the proposed individual centralised critic approach for Mac-IAICC significantly improves upon this.

**Summary Of The Review:**

The main contribution is relatively interesting and contains some novelty. It is a challenging domain and there is a clear industrial application for those that can get this right. I think there are some weaknesses in explanations and definitions in the main paper which weaken it though. The interpretation of the  experiments as it relates to macro-action methods is clear and insightful and convinces me that there is value in these ideas. However, the comparison with individual action methods does not acknowledge the high level of manual code/knowledge injection that favours the macro action methods. As such, I am falling below the acceptance threshold for this paper. However, I would encourage the authors to continue with the work as it is interesting and promising.

---

> ### Author Response · Authors · 2021-11-14
> **Response to Reviewer Exb6 (1/2)**
>
> We thank the reviewer’s time and comments.
>
> The reviewer’s understanding is correct, in that,  we are solving the problems modeled as MacDec-POMDPs with predefined macro-actions, where the low-level policy for achieving each macro-action is given and our developed methods focus on learning the high-level policies for agents to make choices over the macro-actions in an asynchronous way.
>
> We would like to mention that having predefined macro-actions in multi-agent problems is a reasonable and feasible assumption that has been demonstrated in several planning-method-based works [C. Amato et al., 2015a, C. Amato et al., 2015b, S. Omidshafiei et al., 2017, N. Hoang et al., 2018]. One essential motivation is that, in real-world multi-robot systems, each robot is already equipped with certain controllers (e.g., a navigation controller, and a manipulation controller), which can be modeled as macro-actions. Hence, a team of robots can collaborate by executing these controllers in an asynchronous way. Recently, Xiao et al. proposed the first set of macro-action-based deep Q-learning methods that enable agents to learn high-level policies. In this work, we aim to fill another gap by proposing the first set of macro-action-based policy gradient methods for multi-agent systems.
>
> The model of a macro-action is general just like the single-agent option case. It is a choice whether to include primitive actions in the macro-action set or not. Like the single-agent case, including one-step macro-actions typically slows learning. In the multi-agent case, it may lead to more complex asynchronicity over agents.
>
> Regarding the observations at different levels, the model is general enough to support using the same or different observation sets at each level. In our case, the information observed at the high-level is the same as the information observed at the low-level, but being maintained at a different frequency so each agent can obtain a low-level observation at every primitive time-step but can only obtain a high-level observation after a macro-action terminates. As our work focuses on learning only high-level policies over predefined macro-actions, the proposed methods do not need low-level observations during training.
>
> We totally agree that the option-discovery technique is an interesting and significant research topic, which is also our long-term goal. In the single-agent case, learning everything for different levels from scratch is already challenging even just in small problems (e.g., the four-room gridworld). It will be more challenging in multi-agent scenarios to learn options from scratch. Therefore, we think the more promising way is to start with leveraging human knowledge such as having predefined macro-actions. Especially, for real-world multi-robot problems, many researchers have developed excellent controllers for navigation and manipulation, it doesn’t make sense to ditch these controllers and let robots learn from the low-level controllers again. Thus, we have to first solve the challenge of learning policies with asynchronous macro-actions, which is the important contribution made by this paper. The frameworks proposed in our paper can be used by researchers in the MARL community as a base to further develop new algorithms that are able to not only learn with predefined macro-actions but also either improve the given macro-actions or discover new macro-actions.
>
> We conduct the comparison between macro-action-based methods and primitive-action-based methods, because we have to show that primitive-action-based methods cannot solve certain multi-agent tasks that can be solved well by using macro-action-based methods.
>
> We will answer the reviewer’s specific questions below, and make further clarifications in our revision:
>
> 1. The $\mathbb H$ is the number of steps until the problem terminates (the horizon).
>
> 2. Yes, in the low-level execution, agents always behave synchronously on primitive-action execution. The asynchronicity of macro-action execution means that agents make high-level decisions over macro-actions at different time steps.
>
> 3. Trajectory squeezing is a filtering process for updating either critics or policies. As we collect agents’ transition tuples at every primitive-step but updating only happens at the moment when macro-action terminates, the squeezing process picks out the transition tuples when either the corresponding agent’s macro-action terminates or a joint macro-action is done, depending on the learning method.
>
> 4. $\vec{h}$ is a joint macro-observation-action history that is maintained by the RNN layer of networks, where a single joint macro-observation-action is created by concatenating all agents’ macro-observation and macro-action vectors.

---

> > ### Author Response · Authors · 2021-11-14
> > **Response to Reivewer Exb6 (2/2)**
> >
> > 5. A joint macro-action $\vec{m} = \times_{i\in I}m_i$ means a joint set of each agent’s macro-action being running at a timestep $t$. A termination of a joint macro-action is defined as when any agent terminates its current macro-action (Xiao., et al, 2019). For example, in a domain with two agents, at timestep $t=0$, $\vec{m}=(m_1^1, m_2^2)$, and then at t=5, agent $2$ terminates its macro-action $m_2^2$ and selects a new macro-action $m_2^3$, while agent $1$ is still under the running of the first macro-action, the joint macro-action at timestpe $t=5$ becomes $\vec{m}=(m_1^1, m_2^3)$
> >
> > With macro-actions, without including primitive actions in the macro-action set we can no longer have the same form of optimality as the primitive action case (just like the single-agent option case). Convergence for at least some of the policy gradient methods can be determined. For example, in the fully centralized learning case: the environment is stationary so that the convergence of the centralized critic and the policy optimization is guaranteed; as the termination of a joint macro-action depending on any agent’s termination, $Q^{\Psi}(\vec{h}, \vec{m})$ is an expectation over possible terminations (in Eq.3 in Xiao et al., 2019), which indicates that asynchronicity will potentially cause high variance in optimization. Also, since they are policy gradient methods, only local optimality that depends on the particular variant can be expected. We will clarify in the document.
> >
> > 6. We are using a linear decaying $\epsilon-$soft policy to further encourage exploration (refer to the Architecture & Training Section in Foerster et al ., 2018 ). We let the $\epsilon$ decay from 1.0 to 0.05/0.01 within a decaying period. During learning, $\Psi(m|h, \epsilon) =(1-\epsilon) \Psi(m|h) + \epsilon \frac{1}{|M|}$. This is a common choice being used (often not explicitly mentioned but applied in implementation) in primitive-action-based multi-agent actor-critic methods, e.g., LIIR(Du et al., NeurlPs 2019), CM3(Yang et al., ICLR 2020), VDAC (Su et al., AAAI 2021), DOP(Wang et al., ICLR 2021).
> >
> > References mentioned above:
> >
> > C.Amato et al., Planning for decentralized control of multiple robots under uncertainty, ICRA, 2015a.
> >
> > C.Amato et al., Policy search for multi-robot coordination under Uncertainty, RSS, 2015b.
> >
> > S.Omidshafiei et al., Semantic-level decentralized multi-robot decision-making using probabilistic macro-observations, ICRA, 2017.
> >
> > N. Hoang et al., Near-optimal Adversarial policy switching for decentralized asynchronous multi-agent systems, ICRA, 2018.
> >
> > Xiao et al., Macro-action-based deep multi-agent reinforcement learning, CoRL, 2019.
> >
> > Foerster et al., Counterfactual multi-agent policy gradients, AAAI, 2018.
> >
> > Du et al., Liir: Learning individual intrinsic reward in multi-agent reinforcement learning, NeurlPs, 2019.
> >
> > Yang et al., Cm3: Cooperative multi-goal multi-stage multi-agent reinforcement learning, ICLR, 2020.
> >
> > Su et al., Value-decomposition multi-agent actor-critics, AAAI, 2021.
> >
> > Wang et al., DOP: Off-policy multi-agent decomposed policy gradients, ICLR, 2021.

---

> > > ### Comment · Reviewer_Exb6 · 2021-11-29
> > > **Changes are an improvement but concerns over fair comparison still exist**
> > >
> > > Hi,
> > >
> > > Thank you for your responses and the changes that you have made do improve the paper. However, my main concern (and one shared by some other reviewers I think) is that the advantage of your approach needs to be more convincingly empirically demonstrated. I think I understand the arguments about there not being an existing macro action method that supports asynchronicity. However, I still think that a) you could and should make it clearly when comparing with primitive level methods that there is an unquantified injection of knowledge due to the manually encoded macro-actions and b) find some way of fairly comparing your methods with prior work. Comparison with existing macro-action methods that must coordinate on initialisation time, might be one way to address this. Alternatively, as suggested by me before, trying to learn macro-actions, or even being given some macro actions and not others, might go some way towards this. As things stand, the authors have not addressed my (and other reviewer's) existing concerns about this work. As such, I have kept my recommendation for weak reject.

---

> > > > ### Author Response · Authors · 2021-11-29
> > > > **Response to Reviewer Exb6**
> > > >
> > > > We thank the reviewer for the further comments.
> > > >
> > > > As there is no prior work considering asynchronicity over agents’ macro-action selections, it is not straightforward how to conduct a comparison with prior work. It is also not clear to us what the reviewer meant by “Comparison with existing macro-action methods that must coordinate on initialisation time”. This does not sound like a straightforward (or fair) comparison. We’d like the reviewer to provide more details about this suggestion.
> > > >
> > > > We will make it more clear in our final version that having macro-actions does require additional knowledge but is very common in many domains such as robotics where we already have controllers for navigation, grasping, etc (as we mentioned in the above response).

---

### Official Review · Reviewer_fiq5 · 2021-11-01

**Correctness:** 3
**Technical Novelty And Significance:** 3
**Empirical Novelty And Significance:** 3
**Recommendation:** 5
**Confidence:** 3

**Main Review:**

•	Strengths:

This paper considers an important problem of asynchronism of action in MARL. The main contribution of this paper is integrating the macro-action-value from the Q-value-based macro-action MARL method into multi-agent policy gradient.

•	Weaknesses:

The reviewer has some concerns of this paper:
1.	What is asynchronism in MARL? Can you define it in this paper? The main concern of this paper is that it introduces the asynchronism of actions in MARL, however, with macro-actions which is built on top of the option in hierarchical RL, this paper mainly introduces a MARL method with macro-action. The main concern is it seems the proposed methods are off-topic
2.	The integration of macro-action-value from previous MARL work is straightforward, the main challenge has been discussed in Xiao et al., 2019. This may make the integration a bit weak. Or can the author highlight the challenge in the paper?
3.	Environmental scenarios evaluated in the experimental section are mainly on hierarchical MARL and multi-task MARL, it hard to find the asynchronism of actions in these scenarios, which is the main contribution of this paper.
4.	There some hierarchical MARL methods [1, 2, 3, 4, 5], and some role-learning [6] methods, can you also compare the proposed methods with some of these methods and discuss these methods?
5.	It seems that related works on asynchronism of actions in MARL is missing.

Reference:

[1] Hierarchical Cooperative Multi-Agent Reinforcement Learning with Skill Discovery

[2] Feudal Multi-Agent Hierarchies for Cooperative Reinforcement Learning

[3] Multi-Agent Common Knowledge Reinforcement Learning

[4] OPtions as REsponses: Grounding Behavioural Hierarchies in Multi-Agent Reinforcement Learning

[5] HAVEN: Hierarchical Cooperative Multi-Agent Reinforcement Learning with Dual Coordination Mechanism

[6] RODE: Learning Roles to Decompose Multi-Agent Tasks



**Summary Of The Paper:**

This paper considers the problem of MacDec-POMDPs where it requires agents to be capable of performing asynchronously without waiting for other agents to terminate. As there is no multi-agent policy gradient method with macro-actions for MacDec-POMDPs, this paper fill this gap and integrates the macro-action-value into multi-agent policy gradient and proposes (i) macro-action-based independent actor-critic (Mac-IAC) method, (ii) macro-action-based centralized actor-critic (Mac-CAC) method, (iii) Naive Independent Actor with Centralized Critic (Naive IACC) as well as Independent Actor with Individual Centralized Critic (Mac-IAICC) via CTDE. Experimental results show that the proposed methods outperforms vanilla baselines.

**Summary Of The Review:**

The writing is ok. This is paper is easy to follow. It would be great if the author can make videos to visualize the learned results presented in the appendix. I would like to increase the score to Week Accept if the authors can address the above concerns during the review session.

---

> ### Author Response · Authors · 2021-11-14
> **Response to Reviewer fiq5**
>
> We thank the reviewer’s time and comments. We will address the reviewer’s main concerns below:
>
> 1. The asynchronicity considered in our paper is that agents select macro-actions and terminate  macro-actions at different time steps. Such asynchronous high-level decision-making over agents is induced by the fact that each macro-action can be executed over different timesteps.  In general, multi-agent fully collaborative problems with macro-actions can be modeled as MacDec-POMDPs. In this paper, we are not introducing any new asynchronicity of actions, instead, we target solving MacDec-POMDPs by proposing the first set of actor-critic policy gradient methods considering the asynchronous macro-action execution over agents when updating macro-action-based (high-level) policies, which has not been done in MARL community.
>
> 2. In macro-action-based multi-agent scenarios, a general challenge is to determine what information to use, when, and how to update agents’ policies to deal with the above asynchronicity. Xiao et al., 2019 tackle this challenge by proposing the first set of macro-action-based Q-learning methods. We aim to fill the other gap that enables agents to learn macro-action-based policies via policy gradient methods. We first use their work as the basis to learn decentralized critics and a centralized critic in our Mac-IAC and Mac-CAC approach respectively, which has not been formulated before. Importantly, the most challenging part (not considered by Xiao et al., 2019) is how to adapt the actor-critic-based CTDE framework to solve MacDec-POMDPs. This challenge comes from the fact that the asynchronicity from a centralized perspective (joint macro-action termination) is totally different with the asynchronicity from a decentralized perspective (each agent’s own macro-action termination), so that naively (Navie Mac-IACC) using a joint macro-action value function (the centralized critic) to update each agent’s decentralized policy is problematic. To solve this, we propose a solution, Mac-IAICC, and empirically demonstrate its advantage over the naive approach.
>
> 3. As our answer to the reviewer’s first concern, the asynchronicity of macro-action execution over agents means that agents select and finish each own macro-action at different time steps. Such asynchronous scenarios exist in all the considered domains. For example, in Fig. 1-b, the blue chef starts with a macro-action Go-Cut-Board-1 and the pink chef starts with a macro-action Get-Plate-1; the blue chef moves left and finishes its macro-action. At this moment, blue needs to make another high-level decision to select next macro-action while pink is still executing Get-Plate-1 as he just moved downwards by one cell and has not reached the plate. The asynchronous high-level decision-making of macro-action selection over agents starts from this moment and will keep happening in agents’ following behaviors. This kind of asynchronicity can be very complex depending on the number of agents and the variable time length for finishing one macro-action, e.g. in Warehouse-B scenario. To better understand, we suggest the reviewer either check the behavior visualization section in Appendix A.4 or watch our re-submitted videos.
>
> 4&5. We thank the reviewer for pointing out these references, but we would like to mention that none of these works involves asynchronous high-level decision-making over agents:
>
> [1] In section 3.4, the authors mention that they synchronize the time points of all agents’ high-level skill choices.
>
> [2] This work is based on Dec-POMDPs with primitive-actions. The proposed method mainly solves the credit assignment problem over agents by having a manager at the high-level to send separate rewards to the low-level workers (agents) for learning each agent's primitive-action-based policy. During execution, agents synchronously execute using primitive actions.
>
> [3] This work considers only Dec-POMDPs rather than MacDec-POMDPs. The hierarchical structure is proposed to maintain different common knowledge at different levels during training. During execution, agents only make synchronous decisions using primitive-actions.
>
> [4] This work enables each agent to learn a probability model over a latent option space of the opponents and incorporate it into a low-level policy computation at every time step. There is no asynchronous high-level decision happening during execution. Also, this work focuses on the competitive case, but our work considers the fully cooperative case.
>
> [5] This work requires all high-level macro-actions to take the exact same $k$ steps, so that there is no asynchronous decision-making over macro-actions.
>
> [6] In this work, the high-level role selector of each agent assigns a role with the same frequency, every $c$ time-steps, so that agents are synchronized at the high-level.
>
> For the above reasons, these approaches cannot be directly used with asynchronous macro-actions and we didn’t consider them as comparisons.

---

> > ### Comment · Reviewer_fiq5 · 2021-11-30
> > **I keep my score**
> >
> > Thanks for providing the clarifications. Based on the overall discussion, I believe that the paper could be considerably improved, so I will keep my score as it is.

---

### Official Review · Reviewer_jJtb · 2021-11-02

**Correctness:** 3
**Technical Novelty And Significance:** 3
**Empirical Novelty And Significance:** 3
**Recommendation:** 6
**Confidence:** 3

**Main Review:**

Strengths:
The paper focuses on a domain that is common in practice but not well studied, and the proposed methods show clear improvements.

Weakness:
The major concern I have is the lack of comparison on other macro-action-based methods (like those mentioned in the third paragraph in section 1). The proposed extensions are natural generalizations from their original forms (which could be a strength), and improvements are expected when compared with methods not designed for the Mac-POMDP.  It would be more interesting to see the comparisons with methods designed for the same domain.

**Summary Of The Paper:**

The paper proposes several extensions of the existing multi-agent independent actor-critic, centralized actor-critic, and independent actor with centralized critic to the MacDec-POMDP for solving multi-agent problems with asynchronous actions. The proposed methods show significant improvements compared with their original forms in several MacDec-POMDP problems.

**Summary Of The Review:**

While the technical contribution and the experiment results are, in my opinion, not super significant, the paper indeed introduces a practical research direction and brings several intuitive first steps towards this direction. Thus, I recommend acceptance of the paper.

---

> ### Author Response · Authors · 2021-11-14
> **Response to Reviewer jJtb**
>
> We thank the reviewer’s time and comments.
>
> Regarding the reviewer’s concern on the lack of comparison, we’d like to mention that this paper is the first work focusing on learning high-level policies over predefined macro-actions via policy gradient in multi-agent domains. There are actually no comparable published baselines. All existing multi-agent RL actor-critic methods that are not designed for MacDec-POMDP, cannot be directly applied with macro-actions, because these methods all rely on the assumption that agents behave synchronously which contradicts the asynchronous nature of multi-agent systems with macro-actions. Moreover, in Fig. 3, we have shown that, with primitive-actions, both fully centralized learning (CAC) and fully decentralized learning (IAC) methods perform quite poorly, and other existing CTDE methods thus are not able to perform better than that.
>
> Regarding the three related works mentioned in the third paragraph of section 1, we didn’t compare with their methods due to the following reasons:
>
> 1. The methods proposed by Xiao et al., 2019 are value-based off-policy learning methods. However, our work is focused on policy gradient methods via on-policy learning. Value-based methods and policy gradient methods have totally different theoretical properties. They can co-exist and are not necessarily comparable, as they can fit well with different sets of tasks. For instance, policy gradient methods are scalable to large and continuous action spaces, unlike value-based methods. This is why we only compare with policy gradient methods and believe both classes of methods have merit.
>
> 2. The method proposed by Menda et al., 2019 considers a typical case that requires an event-driven simulator that considers continuous timing rather than the fixed time-step based simulator used for general multi-agent and single-agent RL problems, making the method not directly comparable.  We aim to propose general multi-agent macro-action-based learning frameworks. Also, Menda et al have not published their code.
>
> 3. Chakravorty et al., 2019 propose to learn both high-level abstraction and low-level policy by extending the option-critic framework into multi-agent problems, rather than depending on predefined macro-actions. The paper is only posted on arXiv and has not been published. Their work only considers much simpler domains since the method needs to learn everything of both levels. The results show that their method even does not have a significant advantage over a primitive-action-based independent actor-critic method (the IAC considered in our paper). Also, there is no result that clearly demonstrates what options are learned by the method. Besides, they have not published the code of the method.
>
> As our paper is the first work to develop macro-action-based multi-agent actor-critic methods, we are in the same situation when the first primitive-action-based IACC framework was posted by Foerster et al., where their COMA framework was only compared with the vanilla IACC and variants of IAC.  We hope to publish our work to a broad audience so that more researchers will join and contribute to this topic by proposing more baseline methods and benchmark macro-action-based multi-agent domains.

---

### Official Review · Reviewer_7V3B · 2021-11-04

**Correctness:** 3
**Technical Novelty And Significance:** 2
**Empirical Novelty And Significance:** 3
**Recommendation:** 5
**Confidence:** 4

**Main Review:**

**Pros**:
* Writing: The writing in the paper was largely clear.
* Experimental setup: The experimental setup was exhaustive and the 3 environments used to evaluate the framework was interesting.

**Cons**:
* Presentation of the Mathematical ideas: In Section 3, the presentation of the framework could have been made clearer. Upon reading the update rule for the policy gradient in the new framework, It was non-trivial to get the intuition behind the formulation. Perhaps some more discussion on the differences between IAC/CAC algorithms to the presented framework could bring out the fundamental changes in a formulation more clearly. I think at least the algorithmic block pertaining to MAC-IAC can be brought to the main paper from the appendix, as it is the first time the new framework is being presented. The following subsections of Section 3, where MAC-CAC and MAC-IACC are introduced can include more intuitive discussions on how each technique differs in their update equations. At the moment, this section is a little difficult to parse.

* Novelty: The presented framework is interesting but still requires additional work in its analysis -- such as the limitations of the gradient updates, sample efficiency of the proposed methods, and uniform framework for analysis of the 3 proposed variants. At the moment, the framework shows potential but can be refined further.

* Baselines used: I felt the baselines used for comparison were limited to the presented techniques and primitive action-based techniques. How do the algorithms presented under primitive actions fair when they are also presented with high-level macro actions in their action set? This might be a fairer comparison to evaluate the technique upon. I am also curious to know how the presented framework performs in an adversarial/competitive setting. Is this framework even applicable in such a setting?

**General Comments**
Overall, the presented ideas show promise in scaling up asynchronous learning of macro-action-based multi-agent policies in cooperative environments. However, at the moment the presentation of the paper can be improved and the ideas presented need to be elaborated upon further. The extension seems only incremental at the moment. The experimental framework presented seems exhaustive in the cooperative setting though.

*Originality*: Moderate

*Clarity*: Moderate

*Quality*: Moderate

*Significance*: Moderate to High

**Summary Of The Paper:**

The paper tackles the problem of learning asynchronous multi-agent policies with macro-actions. The authors present a set of asynchronous multi-agent Actor-Critic methods in order to solve the problem, which allows agents to `directly' optimize policies that are asynchronous and macro-action based. They apply the framework in 3 standard multi-agent learning paradigms: decentralized learning, centralized learning and centralized learning for decentralized execution. They also empirically show the utility of the methods over the standard individual actor-critic method and centralized actor-critic method on 3 multi-agent cooperative tasks: Box pushing, Overcooked and Warehouse.

**Summary Of The Review:**

Overall the paper showed promise in scaling up the learning of asynchronous macro-action-based multi-agent policies. However, it still requires further work in improving the presentation and performing elaborate analysis of the framework on the three settings discussed -- such as limitations of the gradient updates derived. The work seems to be only incremental in comparison to prior work. Hence, I am inclined to reject it at the moment.

---

> ### Author Response · Authors · 2021-11-14
> **Response to Reviewer 7V3B**
>
> We thank the reviewer’s time and comments. We will address the reviewer’s main concerns below:
>
> $\bullet$ Presentation of the Mathematical ideas:
>
> IAC (Foerster et al., 2018) and CAC (Bono et al., 2018) are two standard learning frameworks for MARL. IAC learns an individual critic and individual policy for each agent (Section 2.3), but CAC treats all agents as a big agent and learns a single centralized critic and a single centralized policy, which is then similar to the single-agent actor-critic framework (Section 2.2) that considers joint information over agents. IACC (Foerster et al., 2018) is a standard framework that uses a centralized critic to update each agent’s individual decentralized policy (Section 2.4).
>
> The intuitive differences between Mac-IAC, Mac-CAC and Mac-IACC are similar to the differences mentioned above, but our macro-action-based approaches deal with the challenge of asynchronous macro-action execution over agents. Such a challenge cannot be solved by any existing multi-agent actor-critic methods as they all assume each action takes exactly one timestep.
>
> In Mac-IAC, the updates of each agent’s policy and critic depend on only the corresponding agent’s own macro-action executions (Eq.7).
>
> In Mac-CAC, the updates of the centralized policy and the centralized critic depend on the joint macro-action’s termination (Eq.9).
>
> In Mac-IACC, naively applying the primitive-action-based IACC idea with macro-actions, such as Naive Mac-IACC using a shared centralized critic in all agent’s decentralized policy gradient (Eq.11), leads to an asymmetry problem because the termination of a joint macro-action from the centralized perspective (associated with the centralized critic) can be very different with a macro-action’s termination from each agent’s decentralized perspective. To solve this issue, Mac-IAICC proposes to learn individual centralized critics w.r.t. each agent’s own macro-action terminations and update each agent’s decentralized policy by using its own centralized critic (Eq.12).
>
> We will make these distinctions clearer in our revision. As the intuitive ideas of Mac-IAC and Mac-CAC are straightforward as well as considering the space limitation, we showed their algorithmic blocks in Appendix A.2, and would like to focus on the more challenging and more interesting issue that Mac-IAICC tackles.
>
> $\bullet$ Novelty:
>
> It is not clear to us what the reviewer recommends in order to show the “limitations of gradient updates” and the “sample efficiency of the proposed methods”. We think the former can be interpreted from our results (Fig.3 and Fig.4) that have shown how well each method solves each task, and the latter can also be detected from the learning curves regarding how many episodes each method takes to converge. It is also not clear what the “uniform framework” is expected by the reviewer. If the reviewer could be more explicit, we can provide a more detailed response.
>
> We also would like to mention the novelties and potential impactions of our work as follows:
>
> a) To our best knowledge, this is the first work that proposes macro-action-based actor-critic methods that consider asynchronous high-level decision-making over agents;
>
> b) To our best knowledge, this is the first work that detects and solves a major asymmetric issue when incorporating macro-actions into the traditional CTDE framework;
>
> c)  This work provides a principled manner to generalize primitive-action-based MARL ideas to asynchronous macro-action-based MARL problems;
>
> d) The proposed macro-action-based methods are more scalable and more promising in solving realistic multi-robot RL problems, because in real-world multi-robot systems, robot controllers (e.g., navigating to a place or picking and placing an object) are often modeled as macro-actions and robots execute these controllers asynchronously without waiting for each other’s termination.
>
> $\bullet$ Baselines used:
>
> As we mentioned in the paper and above, the existing multi-agent actor-critic methods all assume synchronous primitive-action executions. These methods cannot directly work with asynchronous macro-actions. Our work proposes principled ways to incorporate macro-actions into a multi-agent actor-critic framework for decentralized learning (Mac-IAC), centralized learning (Mac-CAC), and CTDE (Mac-IAICC).  We also have empirically demonstrated that naively adding macro-actions into the action set of primitive-action-based CTDE actor-critic framework, as Naive Mac-IACC, can not work well in general.
>
> Our work focuses on fully cooperative multi-agent problems modeled as MacDec-POMDPs with a single shared reward function over agents. Extension to competitive problems can be done by designing separate reward functions for each team or each agent.

---

> > ### Comment · Reviewer_7V3B · 2021-12-01
> > **Thanks for the revisions**
> >
> > Thanks for the clarification and revision of the paper! After reading through all the discussions, I will keep my score.

---

### Author Response · Authors · 2021-11-22
**Summary of the Revision**

We thank all reviewers for their time and comments again! We have made the following changes in our revision:

- Clarified the asynchronicity considered in our paper;

- Clarified the definition of the horizon $\mathbb{H}$ for both the Dec-POMDP and the MacDec-POMDP;

- Highlighted the new challenge being addressed in Section 3;

- Revised the sub-section of each proposed method in order to clarify their differences;

- Revised the descriptions of the trajectory squeezing process in Section 2.5 and in Section 3 for each proposed method;

- Provided a visualization of the trajectory squeezing process for each proposed method in Appendix A.2;

- Submitted videos of running macro-action-based policies learned by Mac-IAICC in all domains (detailed explanation in Appendix A.4 ).

---

### Decision · Program_Chairs · 2022-01-20

**Decision:**

Reject

**Comment:**

The paper provides an "asynchronous" method for multi-agent actor-critic with macro-actions. A major contribution of this paper is the integration of the macro-action-value from the Q-value-based macro-action MARL method into multi-agent policy gradient. Although it appears an interesting contribution, reviewers found that several parts of the paper were not clear enough and there is a lack of fair comparison with previous works.